# Frictiotaxis underlies focal adhesion-independent durotaxis

Adam Shellard[1,8], Kai Weißenbruch [1,8], Peter A. E. Hampshire [2,3,8], Namid R. Stillman[1], Christina L. Dix[4], Richard Thorogate[5], Albane Imbert[4], Guillaume Charras [1,5], Ricard Alert [2,3,6,9] ✉ & Roberto Mayor [1,7,9] ✉

Cells move directionally along gradients of substrate stiffness − a process called durotaxis. In the situations studied so far, durotaxis relies on cell-substrate focal adhesions to sense stiffness and transmit forces that drive directed motion. However, whether and how durotaxis can take place in the absence of focal adhesions remains unclear. Here, we show that confined cells can perform durotaxis despite lacking focal adhesions. This durotactic migration depends on an asymmetric myosin distribution and actomyosin retrograde flow. We propose that the mechanism of this focal adhesion-independent durotaxis is that stiffer substrates offer higher friction. We put forward a physical model that predicts that non-adherent cells polarise and migrate towards regions of higher friction − a process that we call frictiotaxis. We demonstrate frictiotaxis in experiments by showing that cells migrate up a friction gradient even when stiffness is uniform. Our results broaden the potential of durotaxis to guide any cell that contacts a substrate, and they reveal a mode of directed migration based on friction. These findings have implications for cell migration during development, immune response and cancer progression, which usually takes place in confined environments that favour adhesion-independent amoeboid migration.

The ability of cells to migrate following environmental gradients underlies many aspects of development, homoeostasis and disease[1,2]. Cells follow gradients in the stiffness of their substrate, a process called durotaxis that has been demonstrated in multiple cell types in vitro and in vivo[3–7]. The prevailing mechanistic view of durotaxis involves the cells' actomyosin machinery producing forces that pull on the underlying substrate through focal adhesions. These pulling forces then bias cell motion in one direction, typically toward the stiffer substrate[5,8–10]. This physical picture, implemented either at the molecular level in clutch models[5] or at the macroscopic level in continuum models[8,9], is supported by experimental evidence, and an essential component of it is the presence of strong cell-substrate adhesions. Whether cells lacking focal adhesions can respond to mechanical gradients is widely acknowledged as a vital question[11–16] that has been largely unaddressed. Its importance is highlighted by the fact that focal adhesion-independent motility is a fundamental mode of migration, often called amoeboid migration, which is classically exhibited by various cancer and immune cells, but can be triggered in practically any cell type by providing 3D confinement[15,17,18].

[1]Department of Cell and Developmental Biology, University College London, Gower Street, London WC1E 6BT, UK. [2]Max Planck Institute for the Physics of Complex Systems, Nöthnitzerst. 38, 01187 Dresden, Germany. [3]Center for Systems Biology Dresden, Pfotenhauerst. 108, 01307 Dresden, Germany. [4]The Francis Crick Institute, 1 Midland Road, London NW1 1AT, UK. [5]London Centre for Nanotechnology, University College London, London WC1H 0AH, UK. [6]Cluster of Excellence Physics of Life, TU Dresden, 01062 Dresden, Germany. [7]Center for Integrative Biology, Faculty of Sciences, Universidad Mayor, Santiago, Chile. [8]These authors contributed equally: Adam Shellard, Kai Weißenbruch, Peter A. E. Hampshire. [9]These authors jointly supervised this work: Ricard Alert, Roberto Mayor. ✉e-mail: ralert@pks.mpg.de; r.mayor@ucl.ac.uk

## Results

### Microchannels with tuneable stiffness

Part of the challenge in addressing whether cells migrating without focal adhesions are capable of responding to stiffness gradients, or indeed the role of substrate stiffness in the context of 3D adhesion-independent motility in general, lies in the technical challenge of fabricating the confined cellular environments of tuneable stiffness that are necessary to study adhesion-independent migration[12]. Focal adhesion-independent motility has commonly been studied using the 'under agarose assay' in which cells emigrate from a free region into a small gap between agarose and glass. However, in this assay, cells must deform their substrate to move and are compressed while migrating. To disentangle the effects of substrate deformation and compression, a newer method uses microchannels fabricated with poly-dimethylsiloxane (PDMS)[19]. However, this material exhibits stiffness in the MPa range, far from physiologically relevant levels of most tissues, and offers little possibility to tune rigidity. We developed different microchannels using a variety of hydrogels (acrylamide/bis-acrylamide and agarose). However, the only one that produced reliable results was the agarose microchannel. Therefore, we describe here an easy-to-use method in which motile and deformable cells spontaneously migrate into and within agarose-surrounded preformed microchannels (Fig. 1a). The microchannels are bonded to a glass coverslip, which was coated with PLL-g-PEG to minimise friction between the cell surface and the glass[20,21], restricting the cell interaction mainly to the agarose. Fabricating microchannels with agarose offers the potential to investigate cells within substrates of physiologically relevant stiffnesses[22]. The dimensions of the channel are determined by a PDMS mould that provides reliable confinement irrespective of microchannel dimensions (Fig. 1b, d, e and Supplementary Fig. 1a–c) and is unaffected by the stiffness of the substrate (Fig. 1d–f and Supplementary Fig. 1d, e). The stiffness of the microchannel is tuned by the concentration of agarose (Fig. 1c). Fluorospheres did not show directed motion within microchannels (Supplementary Fig. 1f, g), confirming that pressure-driven fluid flow was not a factor in this setup. Importantly, stiffness gradients can be created by combining solutions with different agarose concentrations together during microchannel assembly (Fig. 1g, h).

To test our setup as a valid means of assaying confined non-adherent cellular motility, we used a non-adherent subline of Walker 256 carcinosarcoma (henceforth Walker) cells as a well-validated model of a cell type that moves without using focal adhesions[20,23]. The dimension of the microchannels was chosen to closely match the diameter of polarised Walker cells, providing reliable confinement without exerting compression onto the cells (Supplementary Fig. 2a, b). Fluorescent Beads incorporated into the agarose exhibited no displacement in response to cells migrating along the channel walls, indicating that Walker cells do not deform the substrate while migrating (Supplementary Fig. 2c–h and Supplementary Video 1). Although Walker cells are able to attach to fibronectin, they are completely non-adhesive on agarose or glass coated with PLL-g-PEG (Fig. 2a, b and Supplementary Fig. 3). Furthermore, Walker cells introduced into the agarose microchannel setup exhibited classical amoeboid motion characterised by bleb-based rather than lamellipodium-based motility (Fig. 2c, d and Supplementary Video 2), fast migration (Fig. 2e) and lack of focal adhesions (Fig. 2f, g). Together, these observations indicate that Walker cells are non-adhesive and migrate in confinement but not under compression in our agarose microchannel setup.

### Focal adhesion-independent durotaxis

To directly address whether cells are capable of undergoing durotaxis in a focal adhesion-independent manner, agarose microchannels were fabricated to have either uniform stiffness or a stiffness gradient (Fig. 3a). We tracked migratory cells that entered regions of the microchannel which exhibited either uniform or graded stiffness. To quantify persistence in cellular locomotion, we measured the Forward Migration Index[24], here referred to as Durotaxis Index, along the stiffness gradient so that positive values indicate directed motion bias toward stiff substrate, while negative values indicate directed motion bias toward soft substrate. As the entry point of the channels marks the beginning of the cell tracks, all cells initially migrate in one direction (positive values, Fig. 3b–d). However, cells entering substrate of uniform stiffness had a moderate tendency to repolarise and move back, whereas cells entering a stiffness gradient were characterised by much higher persistence as they migrated toward the stiff substrate (Fig. 3b–e and Supplementary Video 3). In addition, cells exhibited a higher migration speed at higher stiffness values (Fig. 3f, Supplementary Fig. 4, and Supplement Video 4). Overall, these results show that Walker cells migrating without focal adhesions can perform durotaxis.

To exclude the possibility that Walker cells use unspecific integrin-based adhesions different from typical streak-like focal adhesions during amoeboid durotaxis, we validated our results in the presence of pharmacological integrin inhibitors. A combination of the cyclic peptide Cilengitide and a blocking antibody targets all β1- or αV-containing integrin heterodimers and therefore inhibits all possible integrin-based adhesions in Walker cells[25–27]. Pre-incubation of an adhesive Walker subpopulation with these drugs prevented cell spreading and completely blocked cell adhesion in most cells (Supplementary Fig. 5). However, this treatment did not affect the durotaxis efficiency of amoeboid Walker cells in microchannels with stiffness gradients (Fig. 4 and Supplementary Video 5). While we observed a moderately higher average migration speed under the inhibitor treatment (Fig. 4c), both treated and untreated Walker cells migrated with high persistence towards the stiffer region, exhibiting the same Durotaxis Index (Fig. 4b, d). Finally, like Walker cells, HL60 neutrophil-like cells, a cell type with a well-described amoeboid behaviour[28], also exhibited biased motion toward stiffer substrate (Supplementary Fig. 6). Together, these results reveal that focal adhesion-independent durotaxis does not require integrins and that it may be a general phenomenon that applies to multiple cell types.

### Amoeboid durotaxis depends on actomyosin flow

It has previously been described that fast amoeboid migration on substrates of uniform stiffness depends on asymmetric myosin distribution and actomyosin retrograde flow[13,16–18,20,29–32]. We therefore asked whether actomyosin retrograde flow was also observed in our cells undergoing focal adhesion-independent durotaxis. Walker cells expressing myosin-GFP were placed in the agarose microchannels with either uniform stiffness or a stiffness gradient, followed by time-lapse imaging. Cells exhibited an accumulation of myosin at the rear and a clear myosin retrograde flow (Fig. 5a–c, Supplementary Fig. 7a–c and Supplementary Videos 6 and 7). Upon cell repolarisation, actomyosin rapidly accumulated at the new cell rear (Fig. 5b, i and Supplementary Video 8). To confirm that myosin was required for cell migration in our experiments, myosin activity was reduced by treating cells with the ROCK inhibitor Y-27632. A clear loss in the rear accumulation of myosin (Fig. 5e), retrograde flow (Supplementary Fig. 7d), bleb formation (Fig. 5f), and impairment of migration (Fig. 5d, g and Supplementary Video 9) was observed in treated cells compared with control cells. Furthermore, higher cell speed correlated with faster retrograde actomyosin flow (Supplementary Fig. 7c) and the strength of the actomyosin gradient, where negative gradients correlated with positive cell velocity and vice versa (Fig. 5j).

Finally, to test if an inversion of the actomyosin concentration profile precedes an event of cell migration reversal, or vice versa, we monitored the peak of the actomyosin intensity field over time in cells that reversed direction (Fig. 5h–k and Supplement Video 8). We measured the peak position in a one-dimensional coordinate system that runs from -1 to 1 and along the cell length, with the cell centre found at

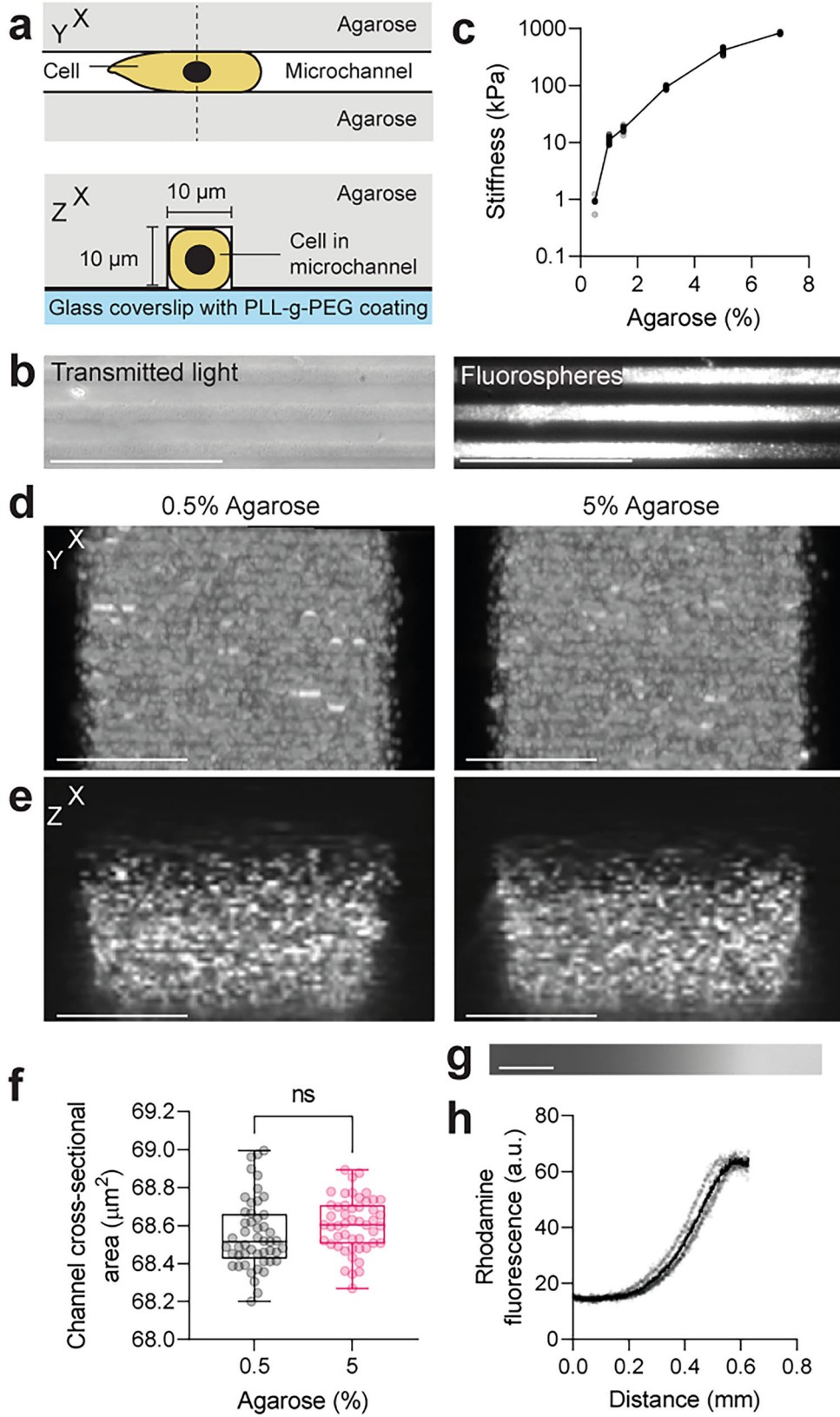

0 (Fig. 5i–k). We found that the actomyosin peak crosses from the back to the front of the cell before the cell reverses direction (Fig. 5k). Together, these observations show that rear accumulation of actomyosin driven by retrograde flow is required for amoeboid migration and determines its direction, suggesting that actomyosin retrograde flow drives durotaxis in our experiments.

**An active gel model of amoeboid migration predicts frictiotaxis**

Given that amoeboid cells lack focal adhesions, it was unclear how these cells were able to durotax. We hypothesised that regions of higher stiffness may offer higher friction. Even in the absence of focal adhesions, cell-wall friction can have multiple origins, including fluid lubrication forces and non-specific molecular interactions between

**Fig. 1 | A microchannel system with tuneable stiffness and dimensions. a** Cross-sectional diagrams of the microchannel assay in XY- and XZ-plane. Microchannel geometry and dimensions of $10 \times 10 \, \mu m$ determine confinement in the setup, agarose concentration determines substrate stiffness. **b** Agarose channels filled with $0.2 \, \mu m$ fluorospheres. Representative image from $N > 3$ independent experiments. Scale bar, $100 \, \mu m$. **c** Stiffness measurements (mean ± s.d. from $N = 3$ experimental repeats) by nanoindentation of gels of different agarose concentration. Grey dots represent individual data points, black dots represent mean. $n = 25$ gels. Agarose microchannels (**d**, **e**; 0.5%, left; 5%, right) filled with $0.2 \, \mu m$ fluorospheres and imaged from below (**d**, maximum projection) and side-on (**e**). Channel cross-sectional area quantified (**f**). Scale bar, $5 \, \mu m$. **f** Box plot for $n = 49$ channels; unpaired two-tailed $t$ test; ns, $P > 0.05$ (exact: $P = 0.2229$). The box bounds the IQR divided by the median, and whiskers extend from minimum to maximum. **g**, **h** A stiffness gradient visualized by immersing rhodamine dextran dye into one of the agarose solutions prior to subsequent diffusion and solidification (**g**) and quantification of the dye fluorescence over the gradient axis (**h**). Scale bar, $100 \, \mu m$. Grey dots represent individual data points, black dots represent mean. $n = 5$ gels. Source data are provided as a Source Data file.

cell-surface proteins like cadherins or glycocalyx components and the channel walls[20,29]. Theoretical models have shown that fast binding-unbinding kinetics of weak cell-substrate bonds leads to a viscous frictional force[33,34] that increases with substrate stiffness[35–37]. Therefore, we hypothesised that, in the absence of adhesion-based active forces pulling the substrate, friction would cause amoeboid cells to undergo durotaxis.

We investigated this hypothesis through a physical model of amoeboid motility which treats the actomyosin cortex as an active gel[13,17,18,20,30–32,38–40]. Following ref. 40, the model is based on a force balance for the actomyosin gel:

$$\xi v = \partial_x \sigma. \tag{1}$$

The left-hand side represents cortex-substrate viscous friction proportional to the coefficient $\xi$ and the cortex velocity $v(x)$ with respect to the substrate. Here, we ignore substrate deformation, because traction stresses exerted by Walker cells migrating in channels have been measured to be around 1 Pa or less[20], which is much smaller than the substrate stiffness of 10–100 kPa that we used. Also in ref. 20, the cell-wall friction coefficient was measured to be in the range $10^3$–$10^7$ Pa s m$^{-1}$ depending on the channel coating. These values of the friction coefficient are orders of magnitude smaller than those for adherent cell migration, obtained for either epithelial cell monolayers[41,42] or keratocyte lamellipodia[43], which in both cases is of the order of $10^9$ Pa s m$^{-1}$.

The right-hand side of Eq. (1) represents the forces generated within the cortex. These forces arise from gradients of the stress $\sigma = \eta \partial_x v + \zeta c$, which includes viscous stresses due to cortex viscosity $\eta$, and active contractile stresses with coefficient $\zeta > 0$ and proportional to the myosin concentration $c(x)$. Myosin is advected by cortical flows and it diffuses with diffusivity $D$, thus satisfying the advection-diffusion equation

$$\partial_t c + \partial_x(cv) = D \partial_x^2 c \tag{2}$$

with no-flux boundary conditions $\partial_x c(\ell_\pm) = 0$ at the cell ends located at $x = \ell_\pm$. The equation and boundary conditions ensure that the total amount of myosin $M = \int_{\ell_-}^{\ell_+} c \, dx$ is conserved. The cell ends move according to the cortical flow: $d\ell_\pm / dt = v(\ell_\pm)$. In addition, changing the cell length $L(t) = \ell_+(t) - \ell_-(t)$ around a reference length $L_r$ is opposed by an elastic force with an effective cell stiffness $k$, which enters as a boundary condition for the stress: $\sigma(\ell_\pm) = -k(L - L_r)/L_r$.

We first consider an unpolarised cell, which has a uniform myosin concentration $c = M/L$ and vanishing cortical flow $v = 0$. In this case, the uniform contractile stress $\sigma = \zeta M/L$ balances the elastic stress at the boundaries to determine the rest length of the cell: $L_{rest} = (L_r + \sqrt{L_r^2 - 4\zeta M L_r/k})/2$. We ignore the other solution for the rest length as it is unstable. Through mechanosensitive mechanisms that are either based on the ion channel Piezo 1[44] or on nuclear shape changes and subsequent calcium release[45,46], confinement, as in our channels, leads to an increased cortical contractility[17,18] $\zeta$. As a result, the rest length decreases and thus the cell contracts (Fig. 6a, left). Moreover, contractility also triggers an instability whereby myosin

accumulates towards one side, which becomes the cell rear[40] (Fig. 6a, right). This concentration profile drives cell motility by contracting the cell rear through the active stress while the elastic stress pushes the cell front forwards. The myosin profile also drives retrograde actomyosin flow, which is consistent with our experiments (Fig. 5a–e).

In the absence of external gradients, the cell contracts equally fast on both sides, and symmetry is broken spontaneously. The cell can thus polarise and move either left or right (Fig. 6a), consistent with our experimental observations of repolarisation events in uniform channels (Fig. 3c, d). In contrast, in the presence of an external friction gradient, the side at lower friction contracts faster than the side at higher friction (Fig. 6b, left). Hence, myosin accumulates faster at the low-friction side, which thus becomes the cell rear (Fig. 6b, right). The cell therefore moves up the friction gradient—a behaviour that we call frictiotaxis.

We demonstrate this mechanism of symmetry breaking in numerical solutions of the model (Methods). We start with an unpolarised cell with initial length $L_0$ longer than the rest length $L_{rest}$. On uniform friction, the cell contracts symmetrically, which leads to an increased myosin concentration in the regions near the boundaries. The increase in myosin intensity is the same at both sides of the cell, so this inhomogeneity then flattens by diffusion. Eventually, the noise in the initial myosin concentration profile triggers the contractile instability and makes the cell move either left or right (Fig. 6c, d and Supplementary Videos 10–11). The final myosin profile is similar to the one in our experiments (Fig. 5e, black). We then consider a friction gradient, which we implement as $\xi(x) = \xi_0 + \xi'(x - \ell_{c,0})$, where $\ell_{c,0}$ is the initial position of the cell centre. We ensure that $\xi'$ is small enough so that the friction is always positive at the position in the cell. On a friction gradient, the cell contracts asymmetrically (Fig. 6e, f and Supplementary Video 12). Again, this leads to myosin accumulation in the regions near the boundaries, but because the flows are faster on the lower-friction side, myosin accumulates more on this side (Fig. 6g). Contractile stresses then drive cortical material to the lower-friction side, which thus becomes the cell rear. Therefore, the velocity of the cell becomes positive (Fig. 6h), indicating that it moves up the friction gradient. Our numerical results therefore showcase that the asymmetric contraction of cells on friction gradients yields frictiotaxis.

Overall, our theory predicts a mode of cell migration guided by friction gradients (Fig. 6b), which we call frictiotaxis. These results reveal a mechanism whereby cells without focal adhesions can perform durotaxis by exploiting gradients in friction rather than in stiffness.

## Substrate friction and stiffness are correlated

To experimentally test the idea that friction is responsible for the observed durotaxis, we first probed the relationship between stiffness and friction by performing lateral force microscopy (LFM) on agarose gels across the stiffness range in which we observed durotaxis. LFM measures the torsional deformation of the micro-mechanical cantilever of an atomic force microscope as it is moved back and forth over the substrate (Supplementary Fig. 8). In our measurements, we functionalised the cantilevers with $10 \, \mu m$ diameter latex beads and imposed a normal force sufficient to obtain contact areas in the $\mu m^2$ range. We then moved the cantilever at speeds between 1 and $10 \, \mu m \, s^{-1}$, of the same order of cell speeds, to measure frictional properties. We

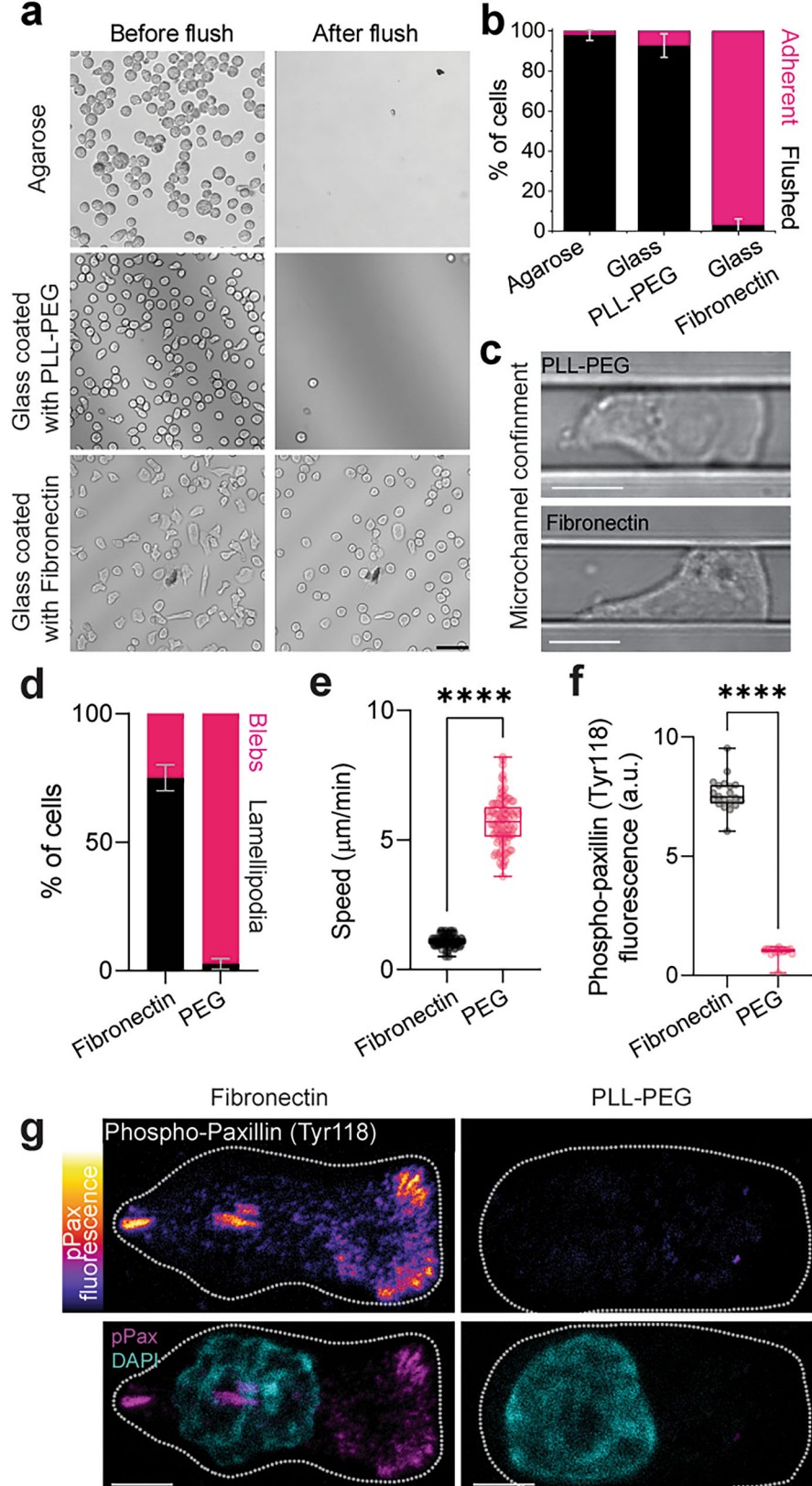

found that, in accordance with our hypothesis, stiffer substrates exhibited greater friction forces (Fig. 7a).

**Experimental evidence of frictiotaxis**

If amoeboid durotaxis is based on friction, it should be impaired if the channel walls are passivated with polyethylene glycol (PEG), which has

been shown to provide low-friction substrates[20,21] (Fig. 7b). Additionally, cells should migrate directionally when stiffness is uniform, but friction is graded.

To test the first prediction, we employed a previously described method in which agarose is coupled to proteins after the surface is activated with cyanogen bromide, enabling functional interactions

**Fig. 2 | Walker cells exhibit adhesion-independent amoeboid migration in the microchannel system. a** Images of Walker cells on agarose, glass coated with PLL-g-PEG, and glass coated with fibronectin, before and after flushing the substrate with culture medium. Scale bar, 50 μm. **b** Quantification of adherent and flushed fraction of cells after flushing the respective substrates. Bars represent mean ± s.d.; $N = 3$ experimental repeats. **c** Example images of cells in non-adhesive (PLL-g-PEG, top) or adhesive (fibronectin, bottom) agarose microchannels. Scale bar, 10 μm. **d** Quantification of lamellipodia and blebs by cells in adhesive (fibronectin) and non-adhesive (PEG) conditions. Bars represent mean ± s.d.; $N = 3$ experimental

repeats. **e** Speed of cells migrating within agarose microchannels with fibronectin or PEG coating. Immunostaining against phospho-paxillin (Tyr118) in Walker cells within agarose channels (**g**) and quantification (**f**) in which the underlying glass is coated with either PLL-g-PEG or fibronectin. Dotted white line represents the cell outline. Scale bar, 5 μm. $n = 100$ cells (**e**, **f**); two-tailed Mann–Whitney test; ****$P ≤ 0.0001$ (exact: $P = 2.25e{-}6$ (**e**) and $6.75e{-}8$ (**f**)). Box plots centres are the median, with bounds representing the IQR and whiskers extending from minima to maxima. Source data are provided as a Source Data file.

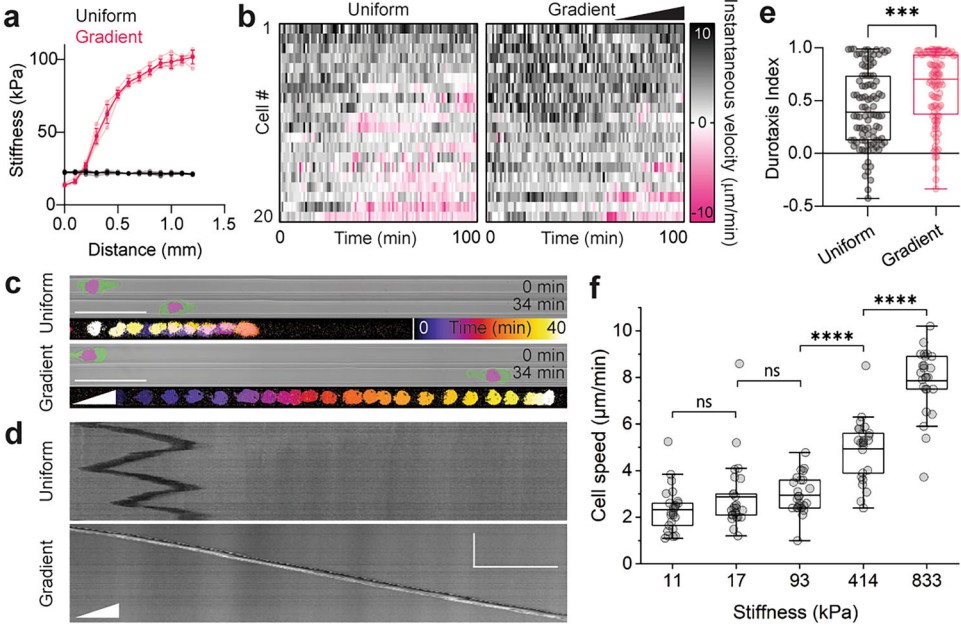

**Fig. 3 | Adhesion-independent durotaxis of Walker cells. a** Stiffness measurements (mean ± s.d.) from nanoindentation of uniform (black) and gradient (pink) microchannels. Opaque dots and lines represent mean; translucent dots and lines represent raw data. $n = 5$ gels each. **b** Heat maps of the instantaneous velocity of representative cells migrating within microchannels of uniform or graded stiffness. $n = 20$ representative cells each. Each row represents a different cell. The x-axis represents time, with each box representing 1 min for a total of 100 min. Each heat map is ordered such that cells migrating forward (black) are at the top, and cells migrating backwards (pink) are at the bottom. **c** Example pictures of cells at an earlier (top panels) and later (middle panels) time point, and temporal colour-coded projected tracks (bottom panels). Cells are tracked by a nuclear marker and

membrane is pseudocoloured. Scale bar, 50 μm. **d** Kymographs of the cells shown in (**c**). Scale bar, 50 μm (horizontal), 50 min (vertical). **e** Quantification of durotaxis. $n = 100$ cells; two-tailed Mann–Whitney test; ***$P ≤ 0.001$ (exact: $P = 0.0003$). **f** Quantification of migration speed in microchannels of different stiffness regimes. $n = 25$ cells for each stiffness regime; two-tailed Mann–Whitney test; ns, $P > 0.05$ (11 versus 17: $P = 0.23258$; 17 versus 93: $P = 0.09795$), ****$P ≤ 0.0001$ (93 vs 414: $P = 2.39e{-}06$; 414 vs 833: $P = 2.42e{-}7$). For (**e**) Box plot centre is the median, with bounds representing the IQR and whiskers extending from minima to maxima. For (**f**) Box plot centre is the mean with bounds representing the IQR and whiskers extending to a maximum of 1.5× IQR beyond the box. Source data are provided as a Source Data file.

---

between the cell and the coated surface[47,48]. As a proof of principle, cells adhered strongly to fibronectin-coated agarose gels (Supplementary Fig. 9a). We thus coated agarose microchannels exhibiting a stiffness gradient with PLL-g-PEG (Supplementary Fig. 9b, c). Cells migrating through such channels had impaired durotaxis, compared to controls (Fig. 7c–e). Thus, durotaxis is impaired by a reduced friction, suggesting that friction enables cells to sense the stiffness gradient.

To address whether gradients in friction are sufficient to guide cell migration in the absence of focal adhesions, we micropatterned gradients of bovine serum albumin (BSA), which provides high friction, and PEG, which provides low friction[20], onto glass and within PDMS microchannels (Fig. 7f–h and Supplementary Fig. 9d, e), such that regions without BSA contained PEG. By performing LFM, we validated that BSA-coated regions provided higher friction than PEG-coated regions (Fig. 7b), which supports previous evidence[20]. Since the length of the gradient is shorter than the overall channel length, several gradients were placed one after the other along each channel. All our assays were carried out in medium without serum, thus avoiding possible binding of serum proteins to BSA that may result in a haptotactic gradient.

We observed cells persistently moving toward areas of higher friction when they migrated in these channels, as opposed to randomly directed motion when friction was uniform (Fig. 7h–j and Supplementary Video 13). Motion guided by friction gradients was previously conceptualized[49] and demonstrated in magnetic colloidal particles[50] and in the contraction of actomyosin networks[51]. Here, our experimental data confirm that cells perform frictiotaxis, and they suggest that this mode of directed migration provides the mechanism for focal adhesion-independent durotaxis. Together, these experiments show that friction gradients guide amoeboid migration—a behaviour that we called frictiotaxis.

## Discussion

Altogether, our study contributes to the growing field questioning how the physical environment modulates focal adhesion-independent migration[15,31,52,53]. Our results reveal that strong and specific adhesions are not required for durotaxis thanks to friction forces, which enable movement toward stiffer substrates because stiffness and friction are correlated. In practice, different tissues may exhibit different stiffness-friction relationships, thus affecting cell response.

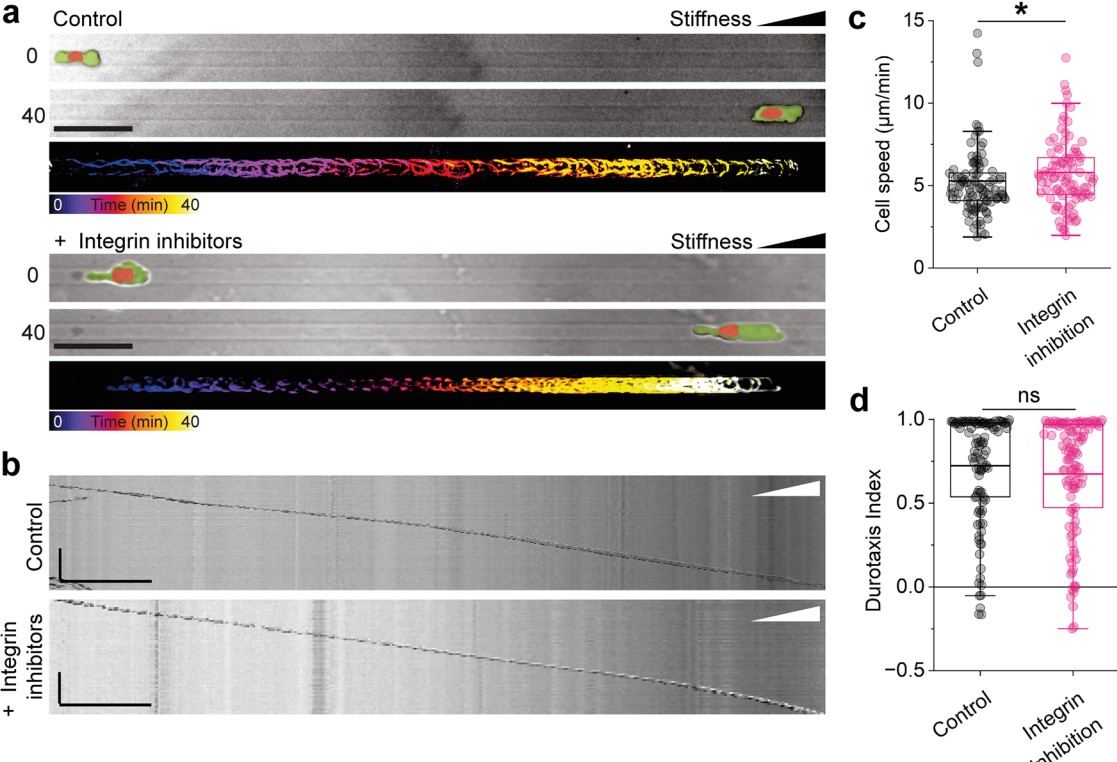

**Fig. 4 | Durotaxis of Walker cells is independent of integrins. a** Example pictures of cells at an earlier (top panels) and later (middle panels) time point, migrating in stiffness gradient channels under control conditions or in the presence of integrin inhibitors (10 µg mL⁻¹ β1-integrin blocking antibody + 10 µM Cilengitide). The cellular membrane and the nucleus are pseudocoloured. Scale bar, 50 µm. The bottom panels show temporal colour-coded projected tracks of the cell each.
**b** Kymographs of the cells shown in (**a**) over the whole distance. Scale bar, 100 µm

(horizontal), 60 min (vertical). **c** Quantification of cell speed. $n = 103$ cells for control and 98 cells for inhibitor conditions; two-tailed Mann–Whitney test; *$P = 0.01366$. **d** Quantification of durotaxis index. $n = 103$ cells for control and 98 cells for inhibitor conditions; two-tailed Mann–Whitney test; $P = 0.12506$. Box plots centres are the mean with bounds representing the IQR and whiskers extending to a maximum of 1.5× IQR beyond the box. Source data are provided as a Source Data file.

Although cells performing amoeboid migration might have weak adhesions, the mechanism of directed motion that we reveal is entirely based on friction. Accordingly, our model includes no adhesion, i.e. no resistance to normal pulling forces, but just friction that opposes cell-substrate sliding. This model is supported by our experimental observations in which suppression of integrin-based adhesions does not impair frictiotaxis. In a scenario in which weak adhesions were also present, durotaxis based on adhesion and on friction could cooperate.

Despite establishing the dominant role of friction, differential substrate deformation could also contribute to focal adhesion-independent durotaxis, especially on softer substrates, as considered theoretically in a recent study[54]. In this study, we used microchannels that were larger than the nucleus, because amoeboid cells use the nucleus as a mechanical gauge in path-making decisions[55]. We also chose to use pre-formed paths, rather than an 'under agarose assay', to rule out the role of compression on the cells, since cells under compression must physically deform their surroundings to move, which is more difficult when the substrate is stiffer[52]. Whether cell compression and other physical inputs can act as guidance cues for focal adhesion-independent cell motility remains an open question. Notably, however, geometric patterns can guide focal adhesion-independent motility[31,56].

In addition, although we propose a physical mechanism by which amoeboid cells could undergo frictiotaxis, we could not test this mechanism directly. To do this, we would have to observe how cells on a friction gradient polarise through the asymmetric cell contraction predicted by our model. Observing events of cell polarisation is not possible in our current experimental setup as cells are already polarised when they move into the channel. We leave this point for future work.

Our experiments showed that the cells moving up a friction gradient are more persistent than those on uniform friction. Explaining this increased persistence will require an understanding of the mechanism of cell reversals, which remains a challenge for future work. Moreover, whereas we considered the effect of friction gradients imposed externally, another interesting direction for future work is to consider friction gradients generated by the cell[57], for example through deposition of extracellular matrix components[58]. Self-generated friction gradients have also been proposed as a mechanism for collective amoeboid migration[59]. We therefore propose that future experiments could test whether cell clusters can display collective frictiotaxis on externally imposed friction gradients.

Finally, whether frictiotaxis and amoeboid durotaxis are physiologically relevant in the complex in vivo environment, especially considering that chemotactic signals are extremely potent drivers of directional cell migration, is also unknown. Confinement promotes bleb-based amoeboid migration[16]. Thus, cell types that naturally undergo rapid amoeboid migration in 3D environments are the most likely candidates to perform frictiotaxis and amoeboid durotaxis. One such candidate are immune cells, which often need to traverse tissues of high density to reach target sites. Indeed, T cells, neutrophils and *Dictyostelium* were recently found to undergo durotaxis with reduced adhesion[15], although whether they are able to sense a friction gradient remains to be investigated. In addition, one of the first responses upon wounding is the recruitment of platelets that exert contractile forces to stiffen fibrin matrices, facilitating subsequent stages of healing[60,61]. This stiffening could provide a gradient in friction, reinforcing the subsequent migration of immune cells towards the wound. Another example with physiological relevance comprises cancer cell migration.

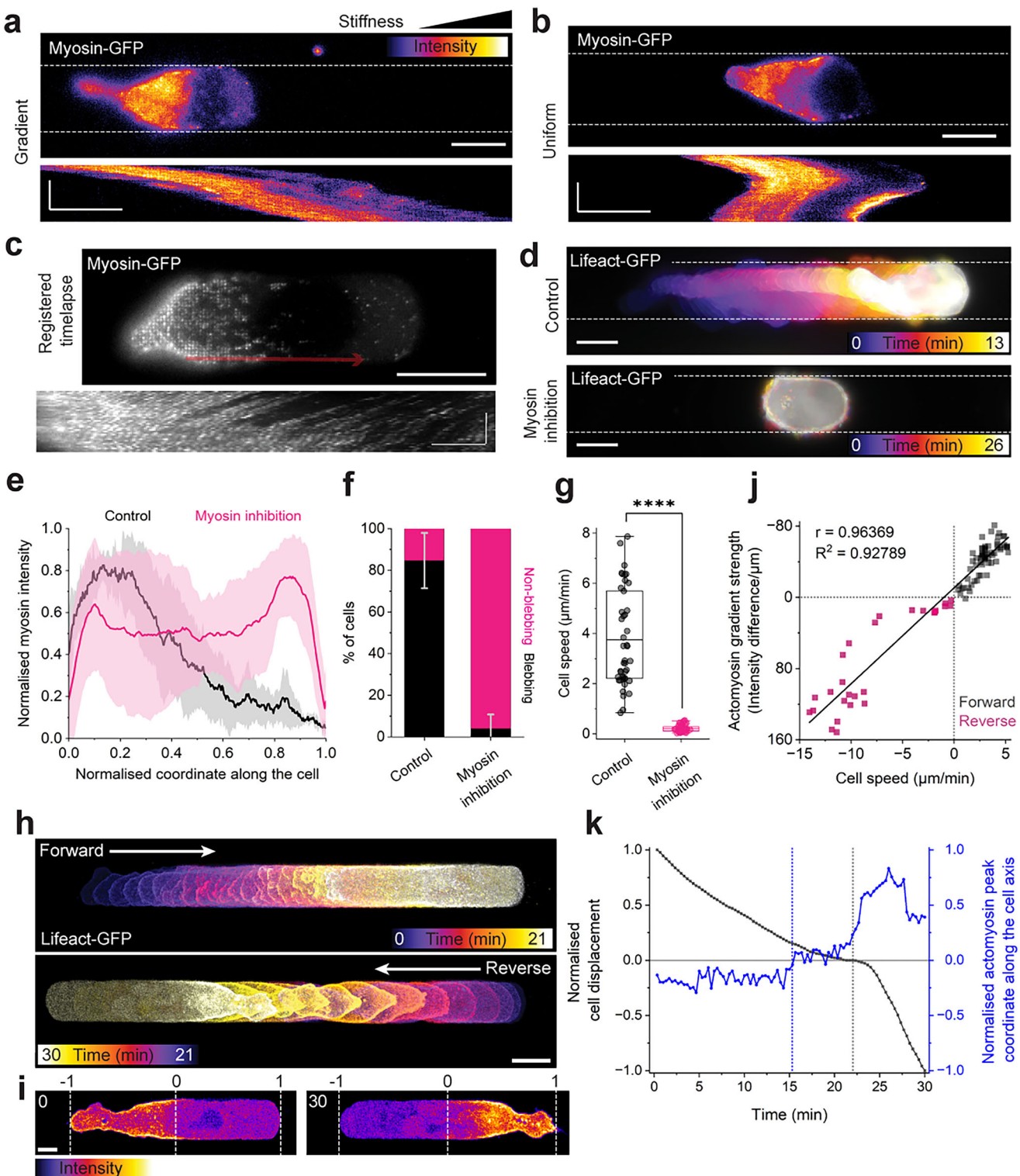

Some types of cancer cells can perform bleb-based migration within the collagen-rich ECM surrounding tumours[62]. Our findings imply that gradients in stiffness of the tumour ECM microenvironment could guide the amoeboid migration of cancer cells. Furthermore, during development, *Drosophila* border cells and zebrafish germ cells migrate under confinement without focal adhesions[63,64], opening the possibility for frictiotaxis. In particular, border cell migration is strongly influenced by substrate topography[65], which suggests that friction could potentially guide border cell migration in vivo. Investigating how focal adhesion-independent durotaxis and frictiotaxis manifest in such intricate, in vivo settings poses an interesting challenge for future work.

## Methods

### Cell culture

Walker 256 carcinosarcoma cells (RRID:CVCL_4984), Walker myosin light chain-GFP cells, and Walker Lifeact-GFP cells were a gift from E. Paluch (University of Cambridge, UK). Walker cells and HL60 cells (RRID:CVCL_0002) were grown in T-25 suspension cell culture flasks (Sarstedt, 83.3910.502) in RPMI 1640 media containing L-glutamine

**Fig. 5 | Adhesion-independent durotaxis depends on retrograde actomyosin flow.** Still images and corresponding kymographs of Myosin-GPF Walker cells in channels with stiffness gradient (**a**) or uniform stiffness (**b**). Fluorescence signal shown as heatmap, dotted lines represent channel. Scale bars, 10 μm in still images, 5 μm (horizontal) and 10 min (vertical) in kymographs. **c** Still image and kymograph (red arrow) demonstrating retrograde myosin flow in migrating Walker cell. Scale bars, 10 μm in still image; 5 μm (horizontal) and 200 s (vertical) in kymograph. **d** Colour-coded projection of Lifeact-GPF Walker cells in uniform stiffness channels, in absence or presence of 30 μM Y-27632. Scale bars, 10 μm. **e** Normalised Myosin-GFP intensity along normalised front-rear axis in absence (black) or presence of 30 μM Y-27632 (purple). Solid lines with transparent areas depict mean ± s.d. of *n* = 6 cells from *N* = 3 independent experiments, each. **f** Blebbing vs non-blebbing cells under control or myosin inhibiting (30 μM Y-27632) conditions. Bars represent mean ± s.d. from *N* = 3 independent experiments.

**g** Migration speed in channels of uniform stiffness. Box plot for *n* = 43 (control) and *n* = 47 (30 μM Y-27632) cells; two-tailed Mann–Whitney test; ****$P \leq 0.0001$ (exact: $P = 4.44e{-}16$). Box bounds the IQR divided by the mean, and whiskers extend to a maximum of 1.5× IQR. **h** Colour-coded projections of Lifeact-GPF Walker cell in channel of uniform stiffness, showing cell reversal event. Scale bar, 10 μm. **i** Cell axis normalisation as shown in (**k**) and actomyosin intensity field (heatmap). Scale bar, 5 μm. **j** Actomyosin gradient strength versus cell speed. Data points obtained from individual frames of Supplement Video 8, as depicted in (**i**). **k** Actomyosin intensity peak along normalised cell axis (blue) versus cell displacement normalised over time (grey). Zero marks cell body centre in cell axis coordinate system. Positive values indicate forward movement, negative values reverse movement on displacement axis. Blue dashed line marks the time when the actomyosin peak crosses the cell axis centre, grey dashed line marks time when cell reverses direction. Source data are provided as Source Data file.

(Gibco, 11875101) supplemented with 10% heat-inactivated FCS and 1% penicillin-streptomycin (Gibco, 10378016) at 37 °C and 5% $CO_2$. The culture media for the HL60 cells additionally contained 40 mM Hepes for buffering. Differentiation of HL60 cells was achieved by incubating cells in propagation media plus 1.3% DMSO for 5 days. All working media was passed through 0.2 μm Sartorius Minisart filters (VWR, 611-0691) prior to use. For experiments, cells were transferred to media identical to their culture media except lacking FCS. For integrin inhibition experiments, media were supplemented with 10 μg mL$^{-1}$ β1-integrin blocking antibody (clone AIIB2, Merck, MABT409) and 10 μM Cilengitide (SigmaAldrich, SML1594). Cells were pre-incubated in this inhibitor mix for 90 min prior to plating. For myosin inhibition experiments, the imaging medium was additionally supplemented with 30 μM of the ROCK inhibitor Y-27632 (Selleckchem, S6390). Cells were allowed to equilibrate in the medium for 30 min before the start of the experiment. An equivalent volume of DMSO was added to the control condition in all inhibitor experiments.

Where appropriate, cells were labelled with Hoechst 33342 (Leica, H3570) at $1 \times 10^{-5}$ mg mL$^{-1}$, or BioTracker 490 Green Cytoplasmic Membrane Dye (Merck, SCT106) at 5 μL per 1 mL cell suspension. In both cases, a concentrated cell solution was incubated with the marker for 30 min at 37 °C before being washed out. Immunostaining against phospho-paxillin Tyr 118 (ThermoFisher, 44-722 G) was performed by fixing in 4% paraformaldehyde for 10 min at 37 °C followed by 0.1% Triton X-100 permeabilization for 15 min at RT, blocking in 2% BSA/PBS for 1 hour at RT, 1:1000 primary antibody overnight at 4 °C and a AlexaFluor goat anti-rabbit secondary antibody (ThermoFisher, A-11008) at 1:500 for 4 hours at RT with PBS washes in between antibody steps.

### Image processing, data analysis and statistics
Images and videos were processed using Fiji. Cells were tracked using the Manual Tracking plugin, and output data processed in the Chemotaxis and Migration Tool (Ibidi, Version 2.0). In this manuscript, we have named Forward Migration Index as the Durotaxis Index, where the axis of interest is along the stiffness gradient and where positive values indicate straightness toward stiff substrate, and negative values indicate straightness toward soft substrate. Further details on the plugin and the full formula for calculation can be found on the Ibidi website (https://ibidi.com/content/306-data-analysis-of-chemotaxis-assays).

To achieve high resolution imaging of cells across large regions, images were stitched for analysis and presentation. Correction for uneven illumination was performed where appropriate. Cells in contact with other cells were not included in the analysis, and tracks were cut short where cell-cell contacts were made, to ensure only single cell analysis was performed.

To quantify retrograde actomyosin flow independently from cell displacement, linear stack alignment with SIFT was carried out in Fiji, and displacement was tracked manually in kymographs, using the segmented line tool.

Normality in the spread of data for each experiment was tested using the Kolmogorov-Smirnov, d'Agostino-Pearson, and Shapiro-Wilk tests in Prism9 (GraphPad9). Significances for datasets displaying normal distributions were calculated in Prism9 with paired or unpaired two-tailed Student's *t*-tests or Mann-Whitney U tests where appropriate. No predetermination of sample sizes was done. Cells were allocated into experimental groups randomly. Cells were selected prior to analysis. Criteria for selection was survival and not interacting with other cells. Therefore, the selection was not blind. All experiments were replicated three times (biological replicates) unless otherwise indicated.

### Nanoindentation
Stiffness measurements were performed using nanoindentation (Chiaro, Optics11Life, Piuma V2 v3.4.3) as previously described[3]. Cantilevers were customized by Optics11 Life. Probes had a spherical glass tip with a radius of ~ 10 μm mounted onto an individually calibrated cantilever with a spring constant of ~0.25 N m$^{-1}$. Deformation of the cantilever after contact with the sample was measured by tracking the phase-shift in light, reflected from the back of the cantilever. Samples were indented to a depth of 0.5 μm with an approach speed of 2.5 μm s$^{-1}$. The tip was held at this indentation depth for 1 s and then retracted over 1 s. The Young's moduli were calculated automatically by the software by fitting the force versus indentation curve to the linear Hertzian contact equation model[66]. The apparent Young's modulus $E$, referred to in this manuscript as stiffness, is derived from the fit of the loading force-displacement curve $F(h)$, the indenter tip radius $R$, and the indentation depth $h$, according to the following formula, for which a Poisson's ratio $v$ of 0.5 was assumed, and was calculated automatically by the software (Chiaro, Optics11Life, Piuma V2 v3.4.3).

$$F = \frac{4}{3}K\sqrt{r\delta^{\frac{3}{2}}} = \frac{4}{3}\frac{E}{1-v^2}\sqrt{r\delta^{\frac{3}{2}}}$$

### Micropatterning
PDMS or glass was plasma treated (Diener electronic) for 2 min and immediately coated with 100 μg/mL PLL(20)-g[3.5]-PEG(2) (Susos) or PLL(20)-g[3.5]-PEG(2)/FITC (Susos), dissolved in PBS, for 30 min at room temperature. Surfaces were washed with 100 mM Hepes pH 8-8.5 for 1 h at room temperature and then incubated with 100 mg/mL mPEG-SVA (Laysan Bio, MPEG-SVA-5000). Surfaces were washed in 1x PBS, followed by water, and then air dried. The photoactivatable reagent, PLPP gel (Alvéole) was added at a ratio of 3:17 with 70% ethanol, at 1 μl PLPP gel/cm², and allowed to dry completely whilst protected from light.

PRIMO (Alvéole) was calibrated with a 20x objective on a Nikon Ti inverted microscope using Leonardo software (Alvéole). A greyscale pattern was used as shown in Supplementary Fig. 9d. A dose of

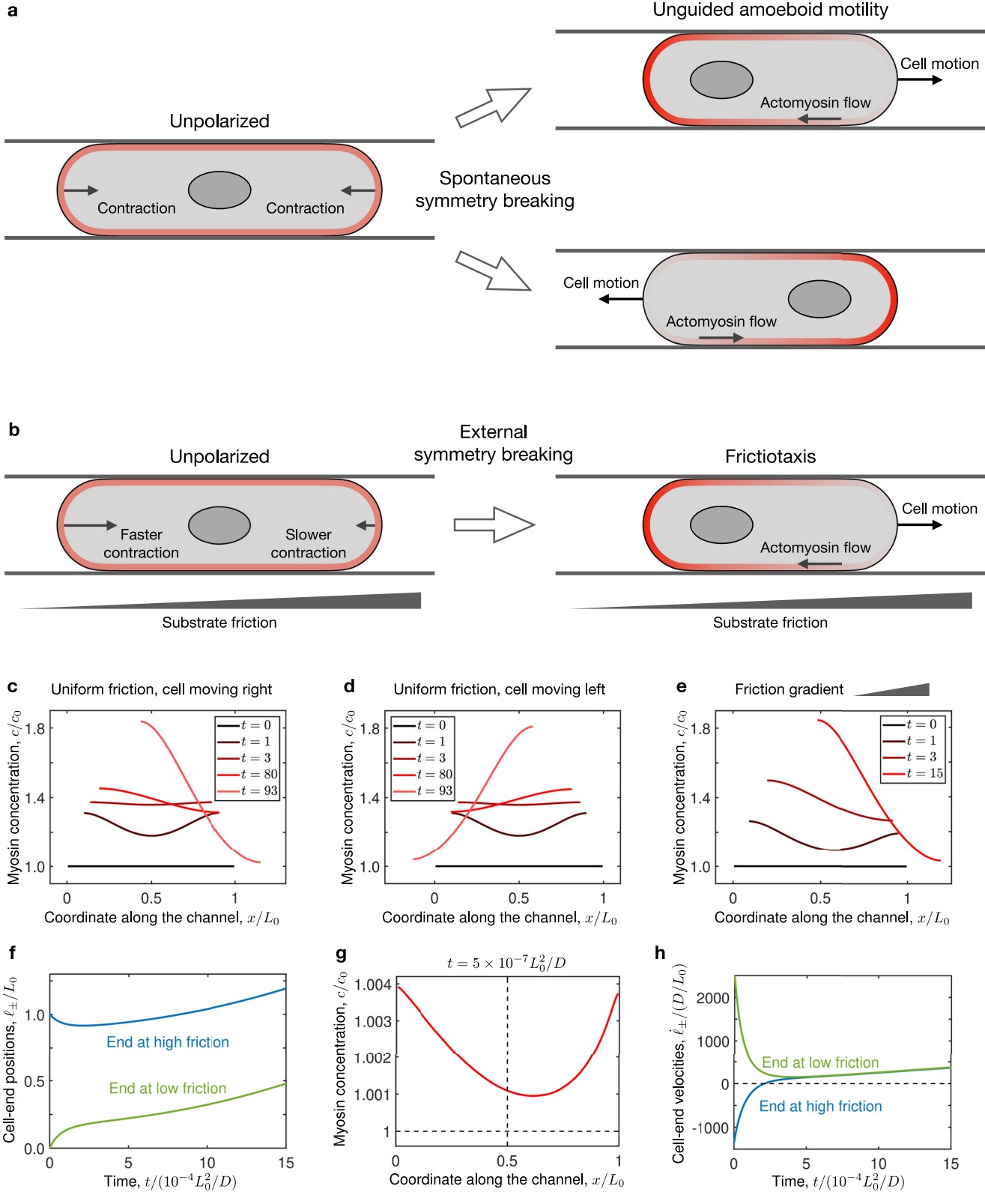

**Fig. 6 | Model of frictiotaxis. a**, **b** Confinement promotes cortical contractility, which produces cell contraction. **a**, In the absence of external gradients, cell contraction is symmetric. Therefore, unpolarised cells break symmetry spontaneously, which leads to amoeboid motility either left or right. **b** On a friction gradient, cell contraction is faster on the lower-friction side, and therefore myosin accumulates faster there. This symmetry-breaking process yields frictiotaxis, that is, directed cell migration towards higher friction. **c–e** Numerical solutions of the model showing how cells polarise by developing asymmetric myosin concentration profiles

(Supplementary Videos 10–12). The time unit in the legends is $10^{-4} L_0^2/D$. On uniform friction, the cell can move either right (**c**) or left (**d**). On a friction gradient, the cell always moves right, i.e., up the friction gradient (**e**). **f–h** Mechanism of symmetry breaking leading to frictiotaxis. The cell contracts asymmetrically (**f**) leading to more myosin accumulation at the lower-friction side at early times (**g**). Active cortical flows then transport myosin to the lower-friction side, which drives cell migration up the friction gradient, shown by a positive cell velocity (**h**). These panels correspond to the solution in (**e**).

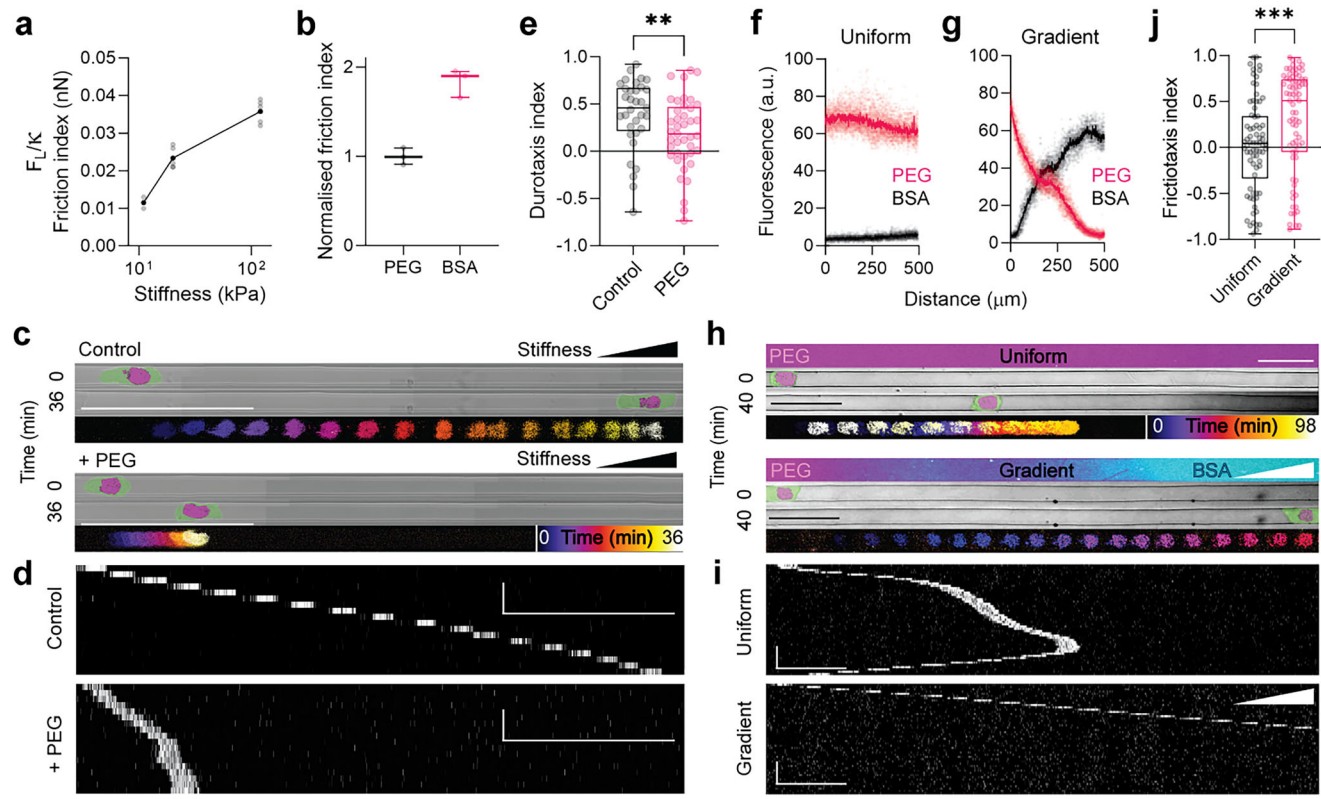

**Fig. 7 | Frictiotaxis in experiments. a** Friction index versus stiffness in the gels; $n = 2$ (in vitro, at 11 kPa); $n = 5$ (in vitro, other stiffness values). Stiffness values correspond to 1% agarose (11 kPa), 2% agarose (20 kPa), and 3% agarose (120 kPa), respectively. **b** Normalised friction index for PEG- or BSA-treated glass (mean ± s.d.); $n = 3$ each. The measurements were normalised to the average value for PEG-coated glass. **c** Pictures of Walker cells with a nuclear label and pseudocoloured membrane after $t = 0$ min and $t = 36$ min in stiffness-gradient microchannels without (Control, upper panel) or with PLL-g-PEG coating ( + PEG, lower panel). Temporal colour-coded projected tracks of the nucleus are shown at the bottom of each of the panels. Scale bars, 100 μm. **d** Kymographs corresponding to the cells shown in (**c**). Scale bars, 100 μm (horizontal) and 10 min (vertical). **e** Durotaxis quantification; $n = 33$ cells; two-tailed Mann–Whitney test; **$P \leq 0.01$ (exact: $P = 0.0097$).

**f**–**g** Quantification of micropatterned PEG and BSA in microchannels with uniform and graded friction; $n = 10$ (PEG); $n = 9$ (BSA). Pale dots represent raw data; dark dots represent mean. **h** Images of PLL-g-PEG/FITC and BSA-Alexa647 micro-patterning (top panels), images of cells with a nuclear marker and pseudocoloured membrane at $t = 0$ min and $t = 40$ min (middle panels), and temporal colour-coded projected tracks (bottom panels). Scale bars, 50 μm. **i** Kymographs corresponding to the cells shown in (**h**). Cells were tracked via the nuclear marker. Scale bars, 50 μm (horizontal) and 10 min (vertical). **j** Quantification of frictiotaxis index from $n = 74$ cells each; two-tailed Mann–Whitney test; ***$P \leq 0.001$ (exact: $P = 0.0010$). Box plots centres are the median, with bounds representing the IQR and whiskers extending from minima to maxima. Source data are provided as a Source Data file.

---

30 mJ/mm² was used and laser power adjusted such that the patterning time for each element was 4 s.

After patterning, the surface was washed several times with water and re-hydrated with PBS for 5 min. The sample was then incubated with 50 μg/mL AlexaFluor647-conjugated BSA (Thermo Fisher, A34785) overnight at 4 °C prior to washing with PBS. Subsequent stages of PDMS bonding to glass and imaging were performed as described in the Soft lithography and Imaging Methods sections.

### Photolithography
3-inch silicon wafers (Silicon wafer test grade, N(Phos), WAFER-SILI-0580W25 from PI-KEM Ltd.) were plasma cleaned for 10 min at 100% power in a plasma cleaner (Henniker Plasma HPT 100), and then baked on a hot plate at 200 °C for 20 min to remove moisture. After 10 s of cooling, SU-8 2005 (Kayaku Advanced Materials, purchased from A-Gas Electronics Materials) was spin coated on the wafer at 500 rpm with an acceleration of 100 rpm s⁻¹ for 30 s. After 2 min rest, the wafer was soft baked at 65 °C for 1 min, followed by 95 °C for 2 min, and then allowed to cool for 10 s. Writing was performed on a MicroWriter ML3 using a 20x objective. Post-exposure bakes of the wafer were then performed at 65 °C for 1 min, 95 °C for 2 min and then 65 °C for 1 min. After 10 s cooling, the wafer was developed in PGMEA (Sigma-Aldrich 484431-1 L) for 1 min with manual agitation, before rinsing in IPA for 10 s, dried with compressed N₂ and hard baked for 20 min at 200 °C.

Dimensions of the wafer were analysed on an optical profiler (Senso-far S Neox).

### Soft lithography
PDMS was made using reagents from a SYLGARD 184 Elastomer Kit (VWR, 634165S). Base elastomer was thoroughly mixed with curing agent at a ratio of a ratio of 10:1, before degassing in a vacuum desiccator (Scienceware, 999320237) with a KNF Laboport N96 diaphragm vacuum pump (Merck, Z675091-1EA). A Si wafer with positive features was placed in a disposable aluminium dish (VWR, 611-1377), covered in PDMS, and baked for 10 min at 110 °C. The PDMS was separated from the wafer and trimmed using a blade.

### Agarose microchannels
PDMS was prepared as above, and spin coated onto a Si wafer with negative features using a spin coater (Spin150) at conditions of 400 rpm for 30 s. The wafer was then baked at 110 °C for 10 min. The thin PDMS was peeled off from the wafer and cut into smaller pieces using a razor blade, where each individual piece contained between one and four patterned designs. To fabricate agarose microchannels, an individual PDMS piece was placed on a glass slide with an edge aligned with the edge of the slide. A vacuum, and subsequently finger pressure, was used to push trapped air bubbles out from underneath the PDMS. A ~1 mm thick piece of PDMS was cut to generate a U-shape

and placed around the PDMS pattern on the slide. A coverslip was placed on top of the U-shape PDMS and neodymium magnets above the coverslip and below the slide were used to secure the sandwich in place.

UltraPure low melting point agarose (ThermoFisher, 16520050)—in this manuscript referred to as 'agarose'—was dissolved in RPMI media at the appropriate concentration, depending on the experiment. The solution was maintained at 70 °C while penicillin-streptomycin (Gibco, 10378016) was added to a final concentration of 1%. To fabricate gels of uniform stiffness, the agarose solution was pipetted into the sandwich until full. To fabricate gels of graded stiffness, higher concentration agarose solution (4%) was pipetted into the sandwich mid-way, followed by lower concentration agarose solution (1%) until full. The gradient was modulated by modifying the diffusion rate, which could be controlled by incubating in an oven at different temperatures, angles and by modifying the input solutions. To solidify the agarose gels, the sandwich was transferred to a 4 °C fridge. Afterwards, the sandwich was disassembled and the solidified agarose slowly slid away from the underlying PDMS mould, and reversed in orientation such that the structured agarose was face up. The surface was then dried with a nitrogen gun and trimmed to size with a razor-blade. The stiffness of agarose gels was measured using a Chiaro nanoindenter ("Methods", nanoindentation).

For Supplementary Figs. Fig. 2c–h, fluorescent beads (Thermo-Fisher Scientific, F8807) were diluted 1:200 in the 1% agarose before casting the channels. Note that due to the porosity of 1% agarose, beads are not stably incorporated in the mesh and might be washed out during the experiment.

Holes were punched using a Harris uni-core 3 mm biopsy puncher (VWR, 89022-356), followed by air drying with a nitrogen gun. A glass coverslip was then plasma cleaned and bonded to the agarose microchannel chip, with the channels facing the coverslip. RPMI media was pipetted into the holes and the unit placed in a humid chamber at 37 °C and 5% $CO_2$ for 30 min. The wells were then emptied using a p200 pipette and replaced with concentrated cell solution. A 0.5 g weighted glass slide was placed on top of the structure, and the unit placed in the stage of an inverted microscope.

Where appropriate, agarose surfaces were covalently bound to protein by using the CNBr-method that has been reported previously[47,48]. After drying, agarose microchannels were activated with cyanogen bromide: 50 mg/mL in water was mixed in an equal ratio with 0.5 M $Na_2CO_3$ in NaOH buffer, pH 11, which contained the protein (fibronectin or PLL-g-PEG). After 30 min, the surface was washed with water and then with the coupling buffer: 0.1 M sodium borate buffer, pH 8.5 for 4 h.

## Measuring the myosin intensity profile

Fluorescent live cell imaging was performed on a ZEISS Elyra7 microscope equipped with a 40×, NA = 1.2 water-immersion objective and operating in laser WF mode, or on an LSM980 operating in the AiryScan 4Y mode and equipped with a 40×, NA = 1.25 water-immersion objective. To prevent GFP quenching, RPMI medium without phenol red (Thermo Fisher, 11835030) was used in all experiments. Time series of single plane images along the cell-agarose interface were acquired in intervals of 10s-20s. We averaged the pixel intensities along the axis perpendicular to the direction of motion to obtain a one-dimensional intensity profile along the cell.

## Lateral force microscopy for estimating substrate friction

**Principle of the measurement.** To provide quantitative measurement of friction, we carried out lateral force measurements using atomic force microscopy[67–70]. The principle of these measurements is as follows. A controlled and constant normal force is applied to a surface via the tip of an AFM cantilever. The cantilever is then dragged back and forth along the surface at a constant speed in a direction perpendicular

to the cantilever axis. Friction between the cantilever tip and the surface creates a torsion of the cantilever that can be measured using the AFM's quadrant photodiode.

**AFM calibration.** We used V-shaped MLCT-B cantilevers with a nominal normal spring constant of 0.02 N/m (Bruker). The normal spring constant of each AFM cantilever was determined using the thermal fluctuation method implemented in the JPK software. The torsional spring constant $k_\theta$ can be calculated from the geometry of the cantilever and the normal spring constant $k_z$ as $k_\theta = \kappa k_z$ with $\kappa$ a constant based on the geometry of the cantilever with units of $m^2/rad$[71]. In our experiments we aimed to compare friction across different conditions and we used cantilevers with identical geometry. Therefore, we report our measurements of lateral force as $F_L/\kappa$.

**LFM protocol.** We reasoned that, to gain informative data about the friction experienced by living cells, we should characterize friction over areas of contact similar to those observed between cells and their substrate. Furthermore, we carried out measurements at speeds comparable with those observed during cell migration. Therefore, we functionalized AFM cantilevers with polystyrene beads with a diameter of 10 μm (Sigma). We then applied normal forces between 0.2 and 1.75 nN, giving an area of contact ~0.5 μm², and we moved the cantilever across the surface at a speed of 2.5 μm/s, over a line of 50 μm. To measure the lateral force, we started cantilever motion and waited until the lateral deflection settled to a constant value (typically taking less than a second). We then estimated the lateral force as $F_L/\kappa = k_z d_L$ with $d_L$ the lateral deflection. We carried out measurements of $F_L/\kappa$ for each surface. This was repeated 5 times for each sample for each speed and each normal force.

We note that our measurements only provide a relative estimate of the friction forces between microbead and substrate. Calculating absolute friction values would require estimation of $\kappa$ as well as validation of our friction force measurements against a substrate of known friction. However, given that we are interested in comparisons across different materials and that we use a microbead as a proxy for cell-surface interactions, we reasoned that these relative estimates are sufficient. We tested the hypothesis that friction varies with stiffness by measuring $F_L/\kappa$ for substrates made from agarose concentrations of 1%, 2%, and 3%, which correspond to stiffnesses of 10 kPa, 20 kPa, and 120 kPa, respectively (Fig. 7a). We also compared the friction of substrates coated with PEG or BSA (Fig. 7b).

**Relationship to the friction force in the theoretical model.** We next provide the relationship between the lateral force $F_L$ and the friction coefficient $\xi$ that we use in our theory. If the cantilever slides at a velocity $V$ over the surface, it experiences a friction force $F_f = \xi V$ on its contact point (Supplementary Figs. Fig. 8). This force produces a torque on the cantilever: $\tau_f = F_f r = \xi V r$, where $r$ is the lever arm as shown in Supplementary Fig. 8. This frictional torque is balanced by the elastic torque arising from cantilever torsion: $\tau_e = k_\theta \theta \approx k_\theta d_L/h$, where $k_\theta$ is the cantilever's torsional stiffness, and $\theta$ is its torsion angle. The torsion angle relates to the lateral displacement $d_L$ as $\sin\theta = d_L/h$, where $h$ is the distance between the cantilever and the detector. For small torsion angles, this relationship can be approximated as $\theta \approx d_L/h$, as used above. The condition of torque balance, $\tau_f = \tau_e$, implies that $\xi V r = k_\theta \theta \approx k_\theta d_L/h$. As shown in the previous paragraph, our lateral force measurement yields the quantity $F_L/\kappa = k_z d_L$. Using the torque balance, together with $k_\theta = \kappa k_z$, we obtain $F_L/\kappa = \xi V r \kappa h$. Therefore, for a given sliding speed $V$, the measured lateral force is proportional to the friction coefficient $\xi$ used in the theory. The proportionality constant involves only geometrical parameters of the cantilever and the experimental setup, which are kept fixed and do not relate to the properties of the substrate. Thus, by measuring the lateral force on different substrates, we probe their friction coefficient $\xi$.

**Numerical solution of the active gel model for amoeboid migration**

To solve the model numerically, we first allowed for non-uniform friction by taking a friction coefficient $\xi(x) = \xi_0 + \xi'(x - \ell_{c,0})$, where $\ell_{c,0}$ is the initial position of the cell centre, and $\xi'$ is the friction gradient. Here, we denote a uniform friction coefficient by $\xi_0$. We then made the equations dimensionless by normalizing lengths by the reference length $L_r$, time by $L_r^2/D$, concentration by $M/L_r$, and stress by $k$, as in Ref. 40, and additionally normalized the friction gradient $\xi'$ by $\xi_0/L_r$. When normalized in this way, the equations have four dimensionless parameters: $\mathcal{Z} = \eta/(\xi_0 L_r^2) = \lambda^2/L_r^2$ compares the screening length $\lambda = \sqrt{\eta/\xi_0}$, at which viscous flows are screened by friction, to the reference length $L_r$; $\mathcal{P} = M\zeta/(kL_r)$ compares the active to the elastic stresses; $\mathcal{K} = k/(\xi_0 D)$ compares the rates of elastic advection to diffusion; and the dimensionless friction gradient $\bar{\xi}' = \xi'/(\xi_0/L_r)$.

We solved the dimensionless equations numerically by using the method in Ref. 72, which is based on the finite-volume method[73]. We first mapped the system of equations to a fixed unit-length domain by using the new spatial coordinate $u = (x - \ell_-)/L$. We discretized the profiles of myosin concentration $c$ and stress $\sigma$ on a grid with $N = 101$ collocation points $u = \mathrm{d}u/2, 3\mathrm{d}u/2, \ldots, 1 - \mathrm{d}u/2$, where $\mathrm{d}u = 1/N$. The velocity was defined on a dual grid with $N + 1 = 102$ collocation points $u = 0, \mathrm{d}u, \ldots, 1$. To discretise the equations, we used centred finite differences to approximate the spatial derivatives, and the friction was evaluated at the central collocation point. Fictitious points were added outside the domain to define the discrete spatial derivatives at the boundaries. We used an explicit Euler step for the equations involving a time derivative with a time step $\mathrm{d}t = 10^{-7}$. The remaining equations were solved algebraically. As initial conditions, we set the cell length to $L(t = 0) \equiv L_0 = L_r$, the cell-centre position to $\ell_c(t = 0) \equiv \ell_{c,0} = L_r/2$, and the value of the myosin concentration at each collocation point to be $c(t = 0) \equiv c_0 = 1$ plus a small Gaussian-noise perturbation with standard deviation $10^{-5}$. The remaining initial values of the myosin concentration, stress and velocity at the collocation and fictitious points were obtained by solving the algebraic equations. We used the parameter values $\mathcal{Z} = 0.05$, $\mathcal{P} = 0.2$ and $\mathcal{K} = 5$, which are similar to the estimates in ref. 40. The friction gradient $\bar{\xi}'$ was set to 0 in Fig. 5c-d (and correspondingly in Supplementary Videos 10 and 11), and to 1.5 for Fig. 5e (and correspondingly in Supplementary Video 12). We checked that the observed mechanism of symmetry breaking and frictiotaxis stays the same when the parameters are varied over several orders of magnitude. This robustness is expected provided that the parameter values are such that the growth rate of concentration perturbations of the uniform state is highest for the longest-wavelength mode, i.e., twice the cell length.

**Reporting summary**

Further information on research design is available in the Nature Portfolio Reporting Summary linked to this article.

## Data availability

All data are available in the main text or the supplementary materials. Source data are provided with this paper.

## Code availability

The codes to solve the model equations and to fit it to the experimental data are provided in the supplementary materials.

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

## Acknowledgements

We thank E. Paluch for gifting us Walker 256 carcinosarcoma cells, L. Alvizi for advice with tissue culture and J. Hartmann for assistance with stitching. We thank P. Saez for advice about PDMS and microchannels. We thank Alveole and Cairn for providing access to the PRIMO, as well as P. March for access to the PRIMO at the University of Manchester. P.H. and R.A. thank P. Recho for providing his code, A. Callan-Jones, P. Haas, J. Neipel, S. Aland, and H.A. Getrack for discussions on the model, and M. Bovyn for discussions on the data analysis. Funding Work in the laboratory of R.M. is supported by grants from the Medical Research Council (MR/S007792/1), Biotechnology and Biological Services Research Council (M008517; BB/T013044) and Wellcome Trust (102489/Z/13/Z). K.W. is supported by the Deutsche Forschungsgemeinschaft (DFG) via the Walter Benjamin Fellowship (513518868).

## Author contributions

A.S. and K.W. conceived the project, performed the experiments, and analysed the experimental data. R.A. and P.H. developed the theory. P.H. produced the numerical solutions and analysed experimental data. N.S. contributed to conceptualisation of the model and performed AFM with R.T. with assistance from G.C. G.C. provided conceptual and technical advice, especially with AFM and tissue culture. A.S. and C.D. generated the silicon wafers in A.I.'s laboratory. A.S., K.W., P.H., R.M. and R.A. wrote and edited the manuscript. All authors commented on the manuscript.

## Competing interests

The authors declare no competing interests.
