## [Transparent Peer Review file · Nature Communications]

Frictotaxis underlies focal adhesion-independent durotaxis

Corresponding Author: Professor Roberto Mayor

Version 0:

Reviewer comments:

Reviewer #1

(Remarks to the Author)

The authors present a compelling experimental and theoretical evidence that confined cells can perform taxis on surfaces with graded friction despite lacking focal adhesions. They conjecture that that stiffer substrates offer higher friction and interpret the observed behavior as durotaxis. The obtained results reveal a new mode of directed migration of cells based on friction. They have implications for cell migration during development, immune response and cancer progression, which usually takes place in confined environments that favor adhesion-independent amoeboid migration. The paper is of high quality and can be published after the authors address the following mostly cosmetic comments:

The author's claim is largely based on the assumption that regions of higher stiffness offer higher friction. However, very little experimental evidence that such rather different properties are correlated is provided. Thus, stiffness is a static feature while friction is a dynamic feature. It was not clear to me what would be the micro mechanism behind the observed correlation? Can the authors propose a micro-model, specifying the molecular agents responsible for friction and linking them to the mechanism of rigidity sensing?

In their modeling the authors rely on the model of viscous friction. In adhesion-dependent models the idea that the time-averaged shear stress is generated by constantly engaging and disengaging focal adhesions is proportional to the velocity of the retrograde flow has a solid theoretical foundation, see for instance, K. Tawada, K. Sekimoto, Protein friction exerted by motor enzymes through a weak-binding interaction. *J. Theor. Biol.* 150(2), 193–200 (1991), but only when the retrograde flow is extremely slow. Can similar analysis be performed for the adhesion-independent model proposed in the paper?

A model of the type used by the authors with space dependent viscous friction coefficient has been previously studied in P. Recho, L. Truskinovsky, Maximum velocity of self-propulsion for an active segment, *Mathematics and Mechanics of Solids* 2016, Vol. 21(2) 263–278. It was assumed there that a cell can control such a distribution actively and even optimize it to achieve maximum efficiency. This raises a question whether in the case of adhesion-independent amoeboid migration the distribution of friction properties is completely passive being induced from outside or it is in fact actively adjusted and even optimized by the cell itself?

Reviewer #2

(Remarks to the Author)

The authors present a manuscript combining theory and experiments around the concept that durotaxis (the motion of biological cells along a substrate stiffness gradient) could be driven by an intermediate variable, the effective friction coefficient between the cell and the substrate in the context of amoeboid motility. To demonstrate the validity of this idea, they first build hydrogel channels with a locally tunable elasticity and then show that certain amoeboid cell types are capable to move in these channels driven by myosin activity. The studied cells are shown to durotax, moving from softer to stiffer regions. An active gel model is then introduced to model this motion and suggests the hypothesis that the sliding friction coefficient is proportional to the substrate stiffness. Such hypothesis is checked with Lateral Force Microscopy. To further demonstrate their idea, the authors then consider the addition of PEG and BSA to modulate the friction independently of the stiffness and indeed observe that the motility is modified according to their prediction.

I find the construction of this manuscript rather compelling. The wording is clear and I particularly appreciate the dialogue between the experiments and the theory. I also think that this idea is novel and important enough in the field of biophysics to warrant publication in *Nature Communication*.

I however have some comments for the authors which I hope could made the manuscript less ambiguous in certain places.

- My most important concern upon reading this manuscript is about the dependence of the friction coefficient on the substrate stiffness. I think a model would help at this level. Can you formulate a -maybe already existing- physical framework that would explain this dependence? In the spirit of this type of works for instance: <https://doi.org/10.1073/pnas.1525462113> ,<https://doi.org/10.1103/PhysRevLett.120.198001>

Concerning the experimental demonstration through LFM can you give references to this method and write a bit more what is done here? I am not a specialist at all but i have heard that the cantilevers of AFMs are difficult to calibrate under this type of lateral contacts. How does the friction index shown on Fig 6 a depend on ξ ?

- This is again a genuinely naïve question of someone who does not know cell culture and whose attention is, maybe wrongly, caught by the term 'serum'. You mention the use of BSA to increase the friction coefficient. It is clear that BSA will not also increase contractility through an enhanced metabolic activity?

The rest of my comments are less important to the validity of the demonstration

-In the introduction you write: "The prevailing mechanistic view of durotaxis involves the cells' actomyosin machinery producing contractile forces that pull on the underlying substrate through focal adhesions." Is it clear that the expansile stresses created by actin polymerization are not also playing an important role in this process?

-At line 42 you are quoting a reference (10) where migration towards a softer part of the substrate is also observed.

-At line 66 you mention that your agarose channels offer a stiffness that is closer to physiological ones compared to PDMS (you show values from 1 to 1000kPa). I certainly agree with this statement but aren't these values still rather far from the ECM stiffness (I have rather 200-300 Pa in mind), the direct environment of the cell?

-Related to the previous point, in eq 1, you may comment that you are considering that the substrate cannot be deformed by the cell, otherwise you would get a substrate velocity entering in this formula. This appears rather justified to me since your softer substrate is of the order of 10kPa in the motility experiments while I would expect cells traction forces to be of the order of 100Pa. But maybe TFM would help to further justify that statement. More generally and as a perspective, It would be interesting to see what happens when the substrate stiffness is in the 0.1-1 kPa range (i guess the lower bound is hard to reach) and the substrate deformation can become important in the presence of TF. Maybe in this case, you have a different regime of durotaxis, not relying on friction but on the substrate deformation.

-when you write "Confinement, as in our channels, leads to an increased cortical contractility" I would include a few words of how this mechanosensing is thought to occur.

- lines 170-171 when you introduce the spatial dependence of ξ , you may need to comment that this expression is OK locally only. When the cell starts to move a finite distance, we expect some saturation/reversal in this expression as it is clear that friction cannot become infinite or negative.

- "Thus, durotaxis is impaired by a reduced friction, suggesting that friction enables cells to produce thrust and migrate." As a comment, the limit of zero friction in your model has been analyzed and still produces a non-zero velocity of the migrating cell (<https://doi.org/10.1103/PhysRevE.105.064401>).

- "Our results reveal that strong and specific adhesions are not required for durotaxis thanks to friction forces, which enable preferential movement toward stiffer substrates because stiffness and friction are correlated." It would be interesting to estimate the friction coefficient of your model and compare it with the friction coefficient inferred (in the group of A Mogliner for instance) for other cell types which migrate with focal adhesion like keratocytes. Is this coefficient indeed considerably smaller?

- "Despite establishing the dominant role of friction, we cannot rule out that differential substrate deformation contributes to focal adhesion-independent durotaxis." We (I suppose it is not very difficult to guess my identity anyway) have recently published a paper that goes in this direction (<https://doi.org/10.1016/j.jmps.2023.105526>). To be operational, this mechanism indeed requires substrates much softer than the ones you use I think.

-It may be interesting for some readers to understand how you modified the existing code to account for a non-constant friction coefficient. Such variations may lead to loss of ellipticity of the screened Poisson equation which may pose some challenge in the choice of the stencil of the numerical scheme.

Reviewer #3

(Remarks to the Author)
Comments for Author

The authors attempt to provide new insights on the ability of cells to migrate along a stiffness gradient, durotaxis (Lo et al., 2000, Bioph Journal). They study the directional migration of a particular type of cells - amoeboid cells that move

independently on the strong cell-substrate adhesions. In most of the established studies, durotaxis relies on cell-substrate focal adhesions to sense stiffness and transmit forces. However, recent studies have shifted focus onto mechanosensing, directional migration and durotaxis of amoeboid cells (Wieland et al., 2021; Kang et al., 2024). With amoeboid cells able to durotax spotted, how durotaxis takes place in the absence of focal adhesions remains an interesting question to be thoroughly addressed. In general, this is an interesting topic and the conclusion from it should benefit fields of immunology and cancer cell biology, since immune cells and multiple cancer cells have been shown to use amoeboid mode for their motility.

The investigators developed an image-based, non-compressive channel to investigate durotaxis of confined cells that do not develop focal adhesions. Subsequently, they used this channel to set up friction gradient, demonstrating gradient of non-specific cell-substrate interaction - as a co-occurrence of stiffness gradient, can induce directional cell migration. I acknowledge their device and methodology design, however, there are three critical issues concerning this manuscript that support my decision of not endorsing this article for publication. See below.

Major points:

First, this stiffness-driven directional migration of amoeboid cells has been recently reported elsewhere (Wieland et al., 2021; Kang et al., 2024). Thus, the phenomenon reported here is not original. The need in the field to explain whether and how cells use amoeboid-like strategies while following a stiffness gradients has a big potential and has not been addressed, indeed this has been previously stated in state of the art reviews (Espina et al., 2022, FEBS). The above mentioned literatures should be taken into account and the authors need to provide more mechanistic insights.

Second, the mechanistic study is not in-depth. The actomyosin flow data the authors provided show rather an association with the more persistent cell movement than the cause of polarized migration in gradient channels. The link between friction and actin flow, and the causal link between flow polarity and cell migration require strong evidence.

Third, the mathematical model is not a predictive tool and as a confirmatory element of their data set. Moreover, the model does not simulate the experimental observations. The authors have stated that as the entry point of the channels marks the beginning of the cell tracks, all cells had a biased tendency to initially migrate in one direction. And cells in uniform channels turn back more often than in gradient channels. However, in their model, the instability is set at the very beginning as a decision for cells to go either left or right. This over simplification significantly weakens the manuscript.

Minor points:

1)The authors claimed that their microchannel device decouples compression vs confinement. This should be demonstrated more carefully with experimental evidence, rather than measuring the parameters of empty channels. The investigators may look into the new preprint from the Legant group (bioRxiv preprint doi: <https://doi.org/10.1101/2024.09.27.615466>)

2)The authors used a kymography to show retrograde actin flow. PIV analysis or similar particle tracking with tools such as u-track should facilitate understanding of the flow field. A very key question is that does the flow direction change prior to cell turning their direction in uniform channels? This is critical since the authors claim that Amoeboid durotaxis depends on actomyosin flow. Whether flow direction is the result or the cause needs to be carefully examined. Currently, the simple pharmacological inhibitor is not sufficient to differentiate. Moreover,

3)Migration persistence is not discussed throughout the manuscript. Is the durotactic behavior shown in this work equivalent to enhanced persistence? Or truly for cells to sense and find their path towards a more stiff end that provide higher friction?

4)Can the authors discuss how do they data relates to the clutch model and known durotactic models that have been previously published, e.g., Sunyer et al Science? Also, I consider important to discuss their data in relation to the discovery of collective durotaxis Sunyer et al Science and recent advances in collective blebbing migration of confined clusters Pagés et al 2022, Sci Advances. This will enrich their discussion.

Reviewer #4

(Remarks to the Author)

The authors have developed an agarose-based 3D microchannel system to investigate the mechanisms of focal-adhesion independent migration of carcinoma cells upwards a gradient of substrate stiffness ("durotaxis"). The model that is investigated does not form focal adhesion complexes in the agarose channels and displays amoeboid migration, based on frontal blebbing and actomyosin contraction at the rear of cells. The authors use stiffness and substrate gradients in the channels, as well as inhibitors of acto-myosin for live-cell imaging of migrating cells. Overall, the data presentation is of highest quality, and the manuscript is exceptionally well written and easy to follow.

A main weakness of the study is that the authors propose the new term "frictotaxis" for the observed phenomena, which I

think this is an overstatement that is not sufficiently supported by the data. All the observations can be simply explained by weak adhesion of cells to the substrate, which may still be mediated through integrins, just not in form of focal-adhesions. Without functional studies that target the adhesion apparatus, either genetically or pharmacologically, no clear conclusion can be drawn about unspecific friction forces at play. For example, the following sentence from the manuscript is a premature statement: "Although cells performing amoeboid migration might have weak adhesions, the mechanism of directed motion that we reveal is entirely based on friction."

Thus, although the data is of high quality, the claim of having discovered a new paradigm in cell migration is not sufficiently supported by the data.

Other comments:

The study primarily investigates amoeboid migration using Walker 256 carcinosarcoma cells, a rather specialized cell type. This raises questions about the physiological relevance of focal adhesion-independent frictotaxis in other cell types.

The theoretical model of asymmetric cell polarization on a friction gradient proposes that the side experiencing lower friction contracts faster, leading to myosin accumulation at the low-friction side, which then becomes the cell's rear. The authors could employ perhaps a form of traction force microscopy to provide evidence supporting this model.

The use of BSA to create friction gradients is also unclear and raises concerns about the potential of BSA to provide chemotactic or haptotactic signals that could influence cell migration.

Version 1:

Reviewer comments:

Reviewer #1

(Remarks to the Author)

The authors adequately answered my concerns and I can now recommend their work for publication in Nature Communications.

Reviewer #2

(Remarks to the Author)

The authors have fully addressed all my concerns. I support publication of this manuscript in its current form.

Reviewer #3

(Remarks to the Author)

After careful revision, the study "Frictotaxis underlies focal adhesion-independent durotaxis" has been significantly improved, with images of adequate quality to support quantification and main conclusions. I believe that in its present form the manuscript is adequate for publication in Nature Communications.

Reviewer #4

(Remarks to the Author)

The authors have provided a thoughtful response to the critiques, addressing the main concerns with new experimental data and clarifications. They effectively demonstrate that their agarose-based system does not support integrin-mediated adhesion, using various controls, including PLL-PEG coating and serum-free conditions. Additionally, they back up their argument by conducting extra experiments with integrin-blocking antibodies and inhibitors, showing that durotaxis efficiency is unaffected. The inclusion of HL60 cells, along with recent findings from Kang et al., further supports the physiological relevance of adhesion-independent migration, expanding the study's impact beyond the Walker 256 model.

However, the introduction of the term "frictotaxis" still raises some questions. While the new data solidify the claim that the observed migration is adhesion-independent, a clearer distinction from other known forms of amoeboid motility would help. The authors should more explicitly contrast frictotaxis with established forms of low-adhesion amoeboid migration to avoid overstating its novelty.

Overall, the manuscript has been significantly strengthened by the authors' responses, and the study presents high-quality data that contribute to the understanding of adhesion-independent migration. Refining the presentation of frictotaxis as a novel concept and exploring its broader implications would enhance the work's impact.

Rebuttal

Reviewer #1

The authors present a compelling experimental and theoretical evidence that confined cells can perform taxis on surfaces with graded friction despite lacking focal adhesions. They conjecture that stiffer substrates offer higher friction and interpret the observed behavior as durotaxis. The obtained results reveal a new mode of directed migration of cells based on friction. They have implications for cell migration during development, immune response and cancer progression, which usually takes place in confined environments that favor adhesion-independent amoeboid migration. The paper is of high quality and can be published after the authors address the following mostly cosmetic comments:

The author's claim is largely based on the assumption that regions of higher stiffness offer higher friction. However, very little experimental evidence that such rather different properties are correlated is provided. Thus, stiffness is a static feature while friction is a dynamic feature. It was not clear to me what would be the micro mechanism behind the observed correlation? Can the authors propose a micro-model, specifying the molecular agents responsible for friction and linking them to the mechanism of rigidity sensing?

The reviewer raises an interesting point here that is very difficult to answer at the current state of knowledge, as any molecule on the cell surface (e.g., glycocalyx components or cadherins) that interacts with the channel wall coating can contribute to cell-wall friction. Our picture is that the cell surface and the channel coating act as two polymer brushes that rub against each other, thus producing friction. The molecules on either surface can transiently bind. Some models showed that when the binding-unbinding kinetics is faster than cortical flows, the cell-substrate force takes the form of viscous friction, with a coefficient that increases with the stiffness of the substrate. These microscopic models, cited below and in the revised manuscript, provide a direct mechanistic link between substrate friction and stiffness.

S. Walcott and S.X. Sun. A mechanical model of actin stress fiber formation and substrate elasticity sensing in adherent cells. *Proc. Natl Acad. Sci. USA* **107**, 7757-7762 (2010).

P. Sens. Rigidity sensing by stochastic sliding friction. *Europhys. Lett.* **104**, 38003 (2013).

Fei et al. Nonuniform growth and surface friction determine bacterial biofilm morphology on soft substrates. *Proc. Natl Acad. Sci. USA* **117**, 7622-7632 (2020).

Finally, we now show in new experiments that integrins (which theoretically could be part of the polymer brushes as well, not just in the form of focal adhesions) are not a necessary component for amoeboid durotaxis in our setup. Which molecular agents instead transduce the retrograde actomyosin flow into friction is a challenging question that needs to be tackled in future work, given the vast and vague list of potential candidate molecules.

*In their modeling the authors rely on the model of viscous friction. In adhesion-dependent models the idea that the time-averaged shear stress is generated by constantly engaging and disengaging focal adhesions is proportional to the velocity of the retrograde flow has a solid theoretical foundation, see for instance, K. Tawada, K. Sekimoto, Protein friction exerted by motor enzymes through a weak-binding interaction. *J. Theor. Biol.* 150(2), 193–200 (1991), but only when the retrograde flow is extremely slow. Can similar analysis be performed for the adhesion-independent model proposed in the paper?*

We thank the reviewer for encouraging us to clarify this point. Indeed, this is the picture that we have of the origin of viscous friction in our system. We now cite the paper mentioned by the reviewer as well as others that have shown this same result based on similar but slightly different models:

K. Tawada and K. Sekimoto. Protein friction exerted by motor enzymes through a weak-binding interaction. *J. Theor. Biol.* **150**, 193-200 (1991).

D. Oriola, R. Alert, and J. Casademunt. Fluidization and Active Thinning by Molecular Kinetics in Active Gels. *Phys. Rev. Lett.* **118**, 088002 (2017).

S. Walcott and S.X. Sun. A mechanical model of actin stress fiber formation and substrate elasticity sensing in adherent cells. *Proc. Natl Acad. Sci. USA* **107**, 7757-7762 (2010).

P. Sens. Rigidity sensing by stochastic sliding friction. *Europhys. Lett.* **104**, 38003 (2013).

Fei et al. Nonuniform growth and surface friction determine bacterial biofilm morphology on soft substrates. *Proc. Natl Acad. Sci. USA* **117**, 7622-7632 (2020).

A model of the type used by the authors with space dependent viscous friction coefficient has been previously studied in P. Recho, L. Truskinovsky, Maximum velocity of self-propulsion for an active segment, Mathematics and Mechanics of Solids 2016, Vol. 21(2) 263–278. It was assumed there that a cell can control such a distribution actively and even optimize it to achieve maximum efficiency. This raises a question whether in the case of adhesion-independent amoeboid migration the distribution of friction properties is completely passive being induced from outside or it is in fact actively adjusted and even optimized by the cell itself?

We thank the reviewer for bringing this paper to our attention, which we cite in the revised manuscript. In our study, we consider the effect of friction gradients imposed externally. We agree with the reviewer that studying the role of cell-generated friction gradients is a fascinating question that we leave for future work.

Reviewer #2

The authors present a manuscript combining theory and experiments around the concept that durotaxis (the motion of biological cells along a substrate stiffness gradient) could be driven by an intermediate variable, the effective friction coefficient between the cell and the substrate in the context of amoeboid motility. To demonstrate the validity of this idea, they first build hydrogel channels with a locally tunable elasticity and then show that certain amoeboid cell types are capable to move in these channels driven by myosin activity. The studied cells are shown to durotax, moving from softer to stiffer regions. An active gel model is then introduced to model this motion and suggests the hypothesis that the sliding friction coefficient is proportional to the substrate stiffness. Such hypothesis is checked with Lateral Force Microscopy. To further demonstrate their idea, the authors then consider the addition of PEG and BSA to modulate the friction independently of the stiffness and indeed observe that the motility is modified according to their prediction.

I find the construction of this manuscript rather compelling. The wording is clear and I particularly appreciate the dialogue between the experiments and the theory. I also think that this idea is novel and important enough in the field of biophysics to warrant publication in Nature Communication.

I however have some comments for the authors which I hope could made the manuscript less ambiguous in certain places.

• My most important concern upon reading this manuscript is about the dependence of the friction coefficient on the substrate stiffness. I think a model would help at this level. Can you formulate a -maybe already existing- physical framework that would explain this dependence? In the spirit of this type of works for instance: <https://doi.org/10.1073/pnas.1525462113>, <https://doi.org/10.1103/PhysRevLett.120.198001>.

We thank the reviewer for encouraging us to clarify this point, which was also raised by Reviewer 1. Lubrication effects could indeed play a role in producing cell-wall friction. In addition to such hydrodynamic effects, our picture is that the cell surface and the channel coating act as two polymer brushes that rub against each other, thus producing friction. The molecules on either surface can transiently bind. Some models showed that when the binding-

unbinding kinetics is faster than cortical flows, the cell-substrate force takes the form of viscous friction, with a coefficient that increases with the stiffness of the substrate. These microscopic models, cited below and in the revised manuscript, provide a direct mechanistic link between substrate friction and stiffness.

S. Walcott and S.X. Sun. A mechanical model of actin stress fiber formation and substrate elasticity sensing in adherent cells. *Proc. Natl Acad. Sci. USA* **107**, 7757-7762 (2010).

P. Sens. Rigidity sensing by stochastic sliding friction. *Europhys. Lett.* **104**, 38003 (2013).

Fei et al. Nonuniform growth and surface friction determine bacterial biofilm morphology on soft substrates. *Proc. Natl Acad. Sci. USA* **117**, 7622-7632 (2020).

• *Concerning the experimental demonstration through LFM can you give references to this method and write a bit more what is done here? I am not a specialist at all but i have heard that the cantilevers of AFMs are difficult to calibrate under this type of lateral contacts.*

We agree with the reviewer that our description of lateral force microscopy was too succinct. We have now expanded the description of LFM experiments in the Methods, including the text that follows together with further details.

To provide quantitative measurements of friction, we carried out lateral force measurements using atomic force microscopy [Ruan, J.-A. & Bhushan, B. 1994; Bhushan et al., 1995]. The principle of these measurements is as follows. A controlled and constant normal force is applied to a surface via the tip of an AFM cantilever. The cantilever is then dragged back and forth along the surface at a constant speed in a direction perpendicular to the cantilever axis. Friction between the cantilever tip and the surface creates a torsion on the cantilever that can be measured using the AFM's quadrant photodiode.

We reasoned that, to obtain informative data about the friction experienced by living cells, we should characterize friction over areas of contact similar to those observed between cells and their substrate. Furthermore, we decided to carry out measurements at speeds comparable with those observed during cell migration. Therefore, we functionalized AFM cantilevers with polystyrene beads with a diameter of 10 μm (Sigma). We then applied normal forces between 0.25 and 1.75 nN, giving an area of contact $\sim 0.5 \mu\text{m}^2$ between the bead and the surface, and moved the cantilever across the surface at speeds between 1 to 10 $\mu\text{m/s}$. To measure the lateral force, we started cantilever motion and waited until the lateral deflection settled to a constant value (typically taking less than a second).

We note that our measurements only provide a relative estimate of the friction forces between the microbead and substrate. Calculating absolute friction values would require estimation of the geometrical factor κ , which relates the torsional stiffness to the normal stiffness of the cantilever. In addition, we would need to validate our friction force measurements against a substrate of known friction. However, given that we are interested in comparisons across different materials and that we use a microbead as a proxy for cell-surface interactions, we reasoned that relative estimates are sufficient.

References:

Matej, G. A., et al. "Precision and accuracy of thermal calibration of atomic force microscopy cantilevers." *Review of Scientific Instruments* 77.8 (2006).

Ruan, Ju-Ai, and Bharat Bhushan. "Atomic-scale friction measurements using friction force microscopy: part I—general principles and new measurement techniques." (1994): 378-388.

Bhushan, B., Israelachvili, J. & Landman, U. Nanotribology: friction, wear and lubrication at the atomic scale. *Nature* **374**, 607–616 (1995).

Blau, Peter J. "The significance and use of the friction coefficient." *Tribology International* 34.9 (2001): 585-591.

Slattery, A. D. et al. Characterisation of the Material and Mechanical Properties of Atomic Force Microscope Cantilevers with a Plan-View Trapezoidal Geometry. *Applied Sciences* 9, 2604 (2019).

How does the friction index shown on Fig 6 a depend on xi?

In our experiments, we measure the lateral deflection of the cantilever as it is dragged across a surface. To compute the lateral force, we need to determine the torsional spring constant of the cantilever. The torsional spring constant k_θ can be calculated from the geometry of the cantilever and the normal spring constant k_z [Slattery et al., *Applied Sciences* 9(13), 2604, (2019)] as $k_\theta = \kappa k_z$ with κ a constant based on the geometry of the cantilever with units of m^2/rad [Slattery et al., *Applied Sciences* 9(13), 2604, (2019)]. In our experiments we aimed to compare friction across different conditions and we used cantilevers with identical geometry. Therefore, we report our measurements of lateral force as F_L/κ , which is given by $F_L/\kappa = k_z d_L$, where d_L is the measured lateral deflection.

We next provide the relationship between the lateral force F_L and the friction coefficient ξ that we use in our theory. If the cantilever slides at a velocity V over the surface, it experiences a friction force $F_f = \xi V$ on its contact point (Extended Data Fig. 8). This force produces a torque on the cantilever: $\tau_f = F_f r = \xi V r$, where r is the lever arm as shown in Extended Data Fig. 8. This frictional torque is balanced by the elastic torque arising from cantilever torsion: $\tau_e = k_\theta \theta \approx k_\theta d_L/h$, where k_θ is the cantilever's torsional stiffness, and θ is its torsion angle. The torsion angle relates to the lateral displacement d_L as $\sin \theta = d_L/h$, where h is the distance between the cantilever and the detector. For small torsion angles, this relationship can be approximated as $\theta \approx d_L/h$, as used above. The condition of torque balance, $\tau_f = \tau_e$, implies that $\xi V r = k_\theta \theta \approx k_\theta d_L/h$. As shown in the previous paragraph, our lateral force measurement yields the quantity $F_L/\kappa = k_z d_L$. Using the torque balance, together with $k_\theta = \kappa k_z$, we obtain $F_L/\kappa = \xi V r \kappa h$. Therefore, for a given sliding speed V , the measured lateral force is proportional to the friction coefficient ξ used in the theory. The proportionality constant involves only geometrical parameters of the cantilever and the experimental setup, which are kept fixed and do not relate to the properties of the substrate. Thus, by measuring the lateral force on different substrates, we probe their friction coefficient ξ .

We have included a new figure (Extended Data Fig. 8) in the revised manuscript to illustrate these aspects.

• *This is again a genuinely naïve question of someone who does not know cell culture and whose attention is, maybe wrongly, caught by the term 'serum'. You mention the use of BSA to increase the friction coefficient. It is clear that BSA will not also increase contractility through an enhanced metabolic activity?*

Although the commercial name indeed contains the word 'serum', BSA solely contains albumin, which is purified from blood serum (hence the name: BSA = bovine serum albumin). BSA is commonly used in cell culture, protein biochemistry, and molecular biology as a blocking agent to prevent unspecific protein binding [Jeyachandran et al., *J. Colloid Sci.* 341(1), 136-142, (2010)]. The most prominent known function of albumins is to maintain the oncotic blood pressure. However, there is no evidence that albumins have any influence on cell contractility and actomyosin activity.

The rest of my comments are less important to the validity of the demonstration

- *In the introduction you write: "The prevailing mechanistic view of durotaxis involves the cells' actomyosin machinery producing contractile forces that pull on the underlying substrate through focal adhesions." Is it clear that the expansile stresses created by actin polymerization are not also playing an important role in this process?*

We thank the reviewer for this question. With the term “contractile”, we referred to the fact that adherent cells exert a contractile traction force dipole on the substrate. These traction forces have multiple contributions, including contractile myosin-generated forces produced in stress fibres as well as retrograde actin flow due to actin polymerization at the cell’s leading edge. To avoid confusion, we have dropped the word “contractile” in this sentence. Regardless of their precise origin, the important point of this sentence is that the cell-generated forces pull on the substrate via focal adhesions.

- At line 42 you are quoting a reference (10) where migration towards a softer part of the substrate is also observed.

Yes, this reference indeed observed durotaxis to both stiffer and softer regions of the substrate. We cite this reference to acknowledge that cells typically — but not always — migrate towards the stiffer region.

- At line 66 you mention that your agarose channels offer a stiffness that is closer to physiological ones compared to PDMS (you show values from 1 to 1000kPa). I certainly agree with this statement but aren't these values still rather far from the ECM stiffness (I have rather 200-300 Pa in mind), the direct environment of the cell?

The stiffness of the ECM depends on many factors like cross-linking density, fibre abundance, fibre alignment, post-translational modifications, etc. All these factors are also influenced by the cells that bind to and remodel the ECM. For reference, we add a figure below showing the physiological stiffness of a variety of tissues [Guimarães et al. *Nature Reviews Materials*. **5**, 351–370, (2020)]. While there are tissues on the lower stiffness regime mentioned by the reviewer, there are also many tissues on stiffness regimes covered by our set-up.

Fig.1: Left panel showing stiffness measurements by nanoindentation of gels of different agarose concentrations, as depicted in Fig.1c in our manuscript. The right panel from Figure 2 in [Guimarães et al. *Nature Reviews Materials*. **5**, 351–370, (2020)] shows the elastic moduli of different tissues. Note that the stiffness of our agarose gels covers the softer regimes of various tissues.

- Related to the previous point, in eq 1, you may comment that you are considering that the substrate cannot be deformed by the cell, otherwise you would get a substrate velocity entering in this formula. This appears rather justified to me since your softer substrate is of the order of 10kPa in the motility experiments while I would expect cells traction forces to be of the order of 100Pa. But maybe TFM would help to further justify that statement.

We thank the reviewer for this comment. We now mention that we ignore substrate deformation because traction stresses exerted on the channel walls by Walker cells have been measured

to be around 1 Pa or less [Bergert et al. *Nat. Cell Biol.* **17**(4), 524-529, (2015)]. This is much smaller than the substrate stiffness in our experiments, which is 10-100 kPa.

In addition, following the suggestion by the reviewer, we performed a new experiment, where we incorporated fluorescent beads into the agarose before casting the gels. These beads exhibited no displacement in response to cells migrating along the channel walls, indicating that Walker cells do not appreciably deform the substrate while migrating (new Extended Data Fig. 2 and new Supplement Video 1).

More generally and as a perspective, it would be interesting to see what happens when the substrate stiffness is in the 0.1-1 kPa range (i guess the lower bound is hard to reach) and the substrate deformation can become important in the presence of TF. Maybe in this case, you have a different regime of durotaxis, not relying on friction but on the substrate deformation.

We agree that this is an interesting question for future work, which is, however, at the moment, not feasible due to technical limitations, as correctly pointed out by the reviewer. At such low stiffness, the integrity of the channel will be compromised.

-when you write "Confinement, as in our channels, leads to an increased cortical contractility" I would include a few words of how this mechanosensing is thought to occur.

We thank the reviewer for encouraging us to clarify this point. Pressure on cells due to confinement is thought to be sensed by Piezo1 channels, which leads to calcium influx and myosin recruitment to the cortex, which triggers a transition to bleb-based amoeboid migration [Srivastava et al. *Proc. Natl Acad. Sci. USA* **117**, 2506, (2020)]. We now mention this mechanosensing mechanism and cite this reference.

- lines 170-171 when you introduce the spatial dependence of ξ , you may need to comment that this expression is OK locally only. When the cell starts to move a finite distance, we expect some saturation/reversal in this expression as it is clear that friction cannot become infinite or negative.

We agree with the reviewer. We now mention this point in the manuscript.

- "Thus, durotaxis is impaired by a reduced friction, suggesting that friction enables cells to produce thrust and migrate." As a comment, the limit of zero friction in your model has been analyzed and still produces a non-zero velocity of the migrating cell (<https://doi.org/10.1103/PhysRevE.105.064401>).

We thank the reviewer for this comment. We have changed the wording of this sentence to "Friction enables cells to sense the stiffness gradient."

- "Our results reveal that strong and specific adhesions are not required for durotaxis thanks to friction forces, which enable preferential movement toward stiffer substrates because stiffness and friction are correlated." It would be interesting to estimate the friction coefficient of your model and compare it with the friction coefficient inferred (in the group of A Mogliner for instance) for other cell types which migrate with focal adhesion like keratocytes. Is this coefficient indeed considerably smaller?

We thank the reviewer for the interesting suggestion. We performed this comparison. For non-adhesive cells in channels, the friction coefficient was measured to be in the range 10^3 - 10^7 Pa s m^{-1} depending on the channel coating [Bergert et al. *Nat. Cell Biol.* **17**(4), 524-529, (2015)]. These values are much smaller than the friction coefficient inferred for cells moving with focal adhesions, obtained either for epithelial cells [Cochet-Escartin et al. *Biophys. J.* **106**, 65-73, (2014), and Rossetti et al. *Nat. Phys.* **20**, 1659-1669, (2024)] or for keratocyte lamellipodia [Rubinstein et al. *Biophys. J.* **97**, 1853-1863, (2009)], which in both cases is around 10^9 Pa s m^{-1} .

- "Despite establishing the dominant role of friction, we cannot rule out that differential substrate deformation contributes to focal adhesion-independent durotaxis." We (I suppose it is not very difficult to guess my identity anyway) have recently published a paper that goes in this direction (<https://doi.org/10.1016/j.jmps.2023.105526>). To be operational, this mechanism indeed requires substrates much softer than the ones you use I think.

We thank the reviewer for the comment. We have cited this paper in our discussion.

- It may be interesting for some readers to understand how you modified the existing code to account for a non-constant friction coefficient. Such variations may lead to loss of ellipticity of the screened Poisson equation which may pose some challenge in the choice of the stencil of the numerical scheme.

We appreciate the suggestion. We have added the details about how the existing code was modified. When the equations are discretised, the friction is always evaluated at the central collocation point. Fortunately, we did not experience any issues related to the loss of the elliptic character of the equation.

Reviewer #3

The authors attempt to provide new insights on the ability of cells to migrate along a stiffness gradient, durotaxis (Lo et al., 2000, Bioph Journal). They study the directional migration of a particular type of cells - amoeboid cells that move independently on the strong cell-substrate adhesions. In most of the established studies, durotaxis relies on cell-substrate focal adhesions to sense stiffness and transmit forces. However, recent studies have shifted focus onto mechanosensing, directional migration and durotaxis of amoeboid cells (Wieland et al., 2021; Kang et al., 2024). With amoeboid cells able to durotax spotted, how durotaxis takes place in the absence of focal adhesions remains an interesting question to be thoroughly addressed. In general, this is an interesting topic and the conclusion from it should benefit fields of immunology and cancer cell biology, since immune cells and multiple cancer cells have been shown to use amoeboid mode for their motility.

The investigators developed an image-based, non-compressive channel to investigate durotaxis of confined cells that do not develop focal adhesions. Subsequently, they used this channel to set up friction gradient, demonstrating gradient of non-specific cell-substrate interaction - as a co-occurrence of stiffness gradient, can induce directional cell migration. I acknowledge their device and methodology design, however, there are three critical issues concerning this manuscript that support my decision of not endorsing this article for publication. See below.

Major points:

First, this stiffness-driven directional migration of amoeboid cells has been recently reported elsewhere (Wieland et al., 2021; Kang et al., 2024). Thus, the phenomenon reported here is not original. The need in the field to explain whether and how cells use amoeboid-like strategies while following a stiffness gradients has a big potential and has not been addressed, indeed this has been previously stated in state of the art reviews (Espina et al., 2022, FEBS). The above mentioned literatures should be taken into account and the authors need to provide more mechanistic insights.

We have considered the literature the reviewer mentioned and referred to in the new manuscript. However, the Wieland et al. (2021) paper addresses a completely different issue, and it never demonstrates or claims that their durotaxis is adhesion-independent; indeed, they use laminin as an adhesive substrate.

More importantly, none of these studies has implemented differential friction in their experimental settings or modelling. Therefore, our finding that a friction gradient is sufficient to guide amoeboid migration under confinement is novel and has not been demonstrated before. This is a striking finding with wide implications for friction-based migration of immune cells or cancer cells under confinement, as also correctly pointed out by the reviewer.

Second, the mechanistic study is not in-depth. The actomyosin flow data the authors provided show rather an association with the more persistent cell movement than the cause of polarized migration in gradient channels. The link between friction and actin flow, and the causal link between flow polarity and cell migration require strong evidence.

The link between friction, actin flow, and cell velocity has already been established in the past using the same cell line as in our study [Bergert et al. *Nat. Cell Biol.* **17**(4), 524-529, (2015)]. This link revealed in previous work underlies friction-based amoeboid migration, and our experiments in Fig. 4 (Fig. 5 in the revised manuscript) show that it also applies to amoeboid durotaxis.

In addition, we have now performed additional experiments and a more in-depth analysis of a cell that reverses direction and upon inversion of its actomyosin concentration profile (included in Figure 5 in the revised manuscript). These new data show that the actomyosin gradient correlates with the migration speed, and an inversion of the actomyosin flow direction precedes an event of cell reversal, significantly strengthening the causal link between actomyosin flow and cell migration.

Third, the mathematical model is not a predictive tool and as a confirmatory element of their data set. Moreover, the model does not simulate the experimental observations. The authors have stated that as the entry point of the channels marks the beginning of the cell tracks, all cells had a biased tendency to initially migrate in one direction. And cells in uniform channels turn back more often than in gradient channels. However, in their model, the instability is set at the very beginning as a decision for cells to go either left or right. This over simplification significantly weakens the manuscript.

We agree that our model does not directly predict the observation in our experiments. The model predicts that unpolarized cells polarize up the friction gradient. As we mention in the discussion, we cannot currently test this prediction directly because we cannot observe cells transition from unpolarized to polarized in our current experimental setup. However, our model captures the main features of amoeboid motility seen in the experiments, such as the myosin concentration profile, which supports the validity of our model. We then showed with numerical solutions that frictiotaxis is possible, and we predicted a mechanism by which it could occur. We believe that this is a novel and exciting result.

In the revised manuscript, we have clarified that the experiments instead showed that the cells moving up a friction gradient have more persistence. Since we do not understand the mechanism of cell reversals, we cannot include this feature in the model to test whether it would capture the observation of increased persistence.

Minor points:

1) The authors claimed that their microchannel device decouples compression vs confinement. This should be demonstrated more carefully with experimental evidence, rather than measuring the parameters of empty channels. The investigators may look into the new preprint from the Legant group (bioRxiv preprint doi: <https://doi.org/10.1101/2024.09.27.615466>)

We thank the reviewer for highlighting this interesting work, which we now cite in our revised manuscript. Unfortunately, 3D TFM is not feasible in our setup, as the porosity of low concentration agarose does not allow the stable incorporation of fluorescent beads to a density that allows deriving displacement fields from it.

We have nevertheless performed additional experiments to address the comment of the reviewer. First, we compared the cellular footprint (diameter of the cell projection against the substrate) of unconfined Walker cells with that of cells in microchannels and “under agarose”. While cells under agarose are significantly compressed and hence show a larger footprint, unconfined cells and cells in microchannels only show slight variations in mean projected diameter, providing evidence that the cells are actually confined but not compressed. Second, we sparsely incorporated fluorescent beads into the agarose and monitored individual beads to observe any possible displacement. However, no displacement was observed, indicating that cells do not deform the substrate during migration. These data are now included in Extended Data Fig. 2 in the revised manuscript.

2) *The authors used a kymography to show retrograde actin flow. PIV analysis or similar particle tracking with tools such as u-track should facilitate understanding of the flow field. A very key question is that does the flow direction change prior to cell turning their direction in uniform channels? This is critical since the authors claim that Amoeboid durotaxis depends on actomyosin flow. Whether flow direction is the result or the cause needs to be carefully examined. Currently, the simple pharmacological inhibitor is not sufficient to differentiate.*

We thank the reviewer for this suggestion. Unfortunately, PIV analysis was not feasible in our movies, but we performed a different analysis to address this question. As seen in the new Figure 5h-k, an inversion of actomyosin flow occurs before a cell reversal event. Together with the previously established link between friction, actin flow, and cell velocity [Bergert et al. *Nat. Cell Biol.* 17(4), 524-529, (2015)], we believe that this is strong evidence for our argumentation.

3) *Moreover, Migration persistence is not discussed throughout the manuscript. Is the durotactic behavior shown in this work equivalent to enhanced persistence? Or truly for cells to sense and find their path towards a more stiff end that provide higher friction?*

We appreciate the comment, and we agree that persistence is the key feature of any tactic behavior. Accordingly, we do talk about persistence. In the section “Focal-adhesion independent durotaxis”, we explicitly introduce the Forward Migration Index (FMI) as a “measure of persistence in cellular locomotion,” and we then state that “cells entering a stiffness gradient were characterised by much higher persistence as they migrated toward the stiff substrate”. Later on, in the section “Experimental evidence of frictiotaxis”, we say “We observed cells persistently moving toward areas of higher friction.”

In practice, as cell migration always has a stochastic component, any guidance mechanism or tactic behaviour acts by increasing the persistence. To further emphasise this point, we now added a new paragraph in the Discussion, where we write “Our experiments showed that the cells moving up a friction gradient have more persistence than those on uniform friction. Explaining this increased persistence will require an understanding of the mechanism of cell reversals, which remains a challenge for future work.”

4) *Can the authors discuss how do they data relates to the clutch model and known durotactic models that have been previously published, e.g., Sunyer et al Science? Also, I consider important to discuss their data in relation to the discovery of collective durotaxis Sunyer et al Science and recent advances in collective blebbing migration of confined clusters Pagés et al 2022, Sci Advances. This will enrich their discussion.*

We thank the reviewer for this question. The clutch is a model of focal adhesions. Therefore, it applies to migration based on focal adhesions, where it was shown to capture durotaxis. Similarly, continuum models that introduce active polarized tractions based on focal adhesions have also captured durotaxis of adherent cells. We have expanded the Introduction to better reflect this point.

The collective blebbing migration in Pagès et al. was suggested to arise from a friction gradient that spontaneously emerges in the cell cluster. We now discuss this point in the Discussion,

and we suggest that future work based on these findings could explore whether such amoeboid cell clusters can display collective frictotaxis on externally imposed friction gradients.

Reviewer #4

The authors have developed an agarose-based 3D microchannel system to investigate the mechanisms of focal-adhesion independent migration of carcinoma cells upwards a gradient of substrate stiffness (“durotaxis”). The model that is investigated does not form focal adhesion complexes in the agarose channels and displays amoeboid migration, based on frontal blebbing and actomyosin contraction at the rear of cells. The authors use stiffness and substrate gradients in the channels, as well as inhibitors of acto-myosin for live-cell imaging of migrating cells. Overall, the data presentation is of highest quality, and the manuscript is exceptionally well written and easy to follow.

A main weakness of the study is that the authors propose the new term “frictotaxis” for the observed phenomena, which I think this is an overstatement that is not sufficiently supported by the data. All the observations can be simply explained by weak adhesion of cells to the substrate, which may still be mediated through integrins, just not in form of focal-adhesions. Without functional studies that target the adhesion apparatus, either genetically or pharmacologically, no clear conclusion can be drawn about unspecific friction forces at play. For example, the following sentence from the manuscript is a premature statement: “Although cells performing amoeboid migration might have weak adhesions, the mechanism of directed motion that we reveal is entirely based on friction.” Thus, although the data is of high quality, the claim of having discovered a new paradigm in cell migration is not sufficiently supported by the data.

We appreciate the reviewer’s comment. However, we want to again stress that our amoeboid durotaxis assay is carried out under strictly non-adhesive conditions:

- Agarose is inert and doesn’t adsorb proteins that might serve as integrin ligands (Fig. 2a).
- The glass coverslip underneath the agarose is coated with PLL-PEG, which also prevents protein adsorption and integrin-based adhesion (Fig. 2a).
- The cell culture medium does not contain any FCS, preventing integrin adhesion via adsorption of soluble factors in the serum to the glass or agarose.

To nevertheless rule out any possible contribution of integrins via unspecific adhesion to the substrate, we performed a series of new experiments. We incubated Walker cells with a combination of an established β 1-integrin blocking antibody (Sigma-Aldrich #MABT409) and the pharmacological α V-integrin inhibitor Cilengitide [Mas-Moruno et al. *Anticancer Agents Med Chem.* **10**(10), 753-768, (2010)]. This combination inhibits all possibly relevant heterodimers; the only integrins that are not targeted by this strategy are leukocyte- or platelet-specific and therefore do not play a role in our cells [Hynes. *Cell* **110**(6), 673-687, (2002)]. As expected, this treatment strongly inhibits cell adhesion of an adherent Walker cell subpopulation (Extended Data Fig. 5 in the revised manuscript). However, no difference in durotaxis efficiency was observed, when we compared control cells and cells incubated with the inhibitor combination (Figure 4 in the revised manuscript).

Taken together, we believe that these are strong arguments, justifying our conclusion that frictotaxis is based on unspecific interactions between the cell membrane and the substrate. In addition, our theory, which includes no adhesion, shows that cells can follow a friction gradient. Showing that frictotaxis can exist and providing strong experimental evidence of it is a novel and relevant finding.

Other comments:

The study primarily investigates amoeboid migration using Walker 256 carcinosarcoma cells, a rather specialized cell type. This raises questions about the physiological relevance of focal adhesion-independent frictotaxis in other cell types.

In addition to Walker cells, we also show amoeboid durotaxis of HL60 cells (Extended Data Fig. 6 in the revised manuscript). Moreover, while this manuscript was under review, another study by Kang et al. was published [Kang et al. *eLife* **13**: RP96821, (2024)], showing in a different set-up that also primary T-cells, neutrophils, and *Dictyostelium* undergo adhesion-independent durotaxis. This highlights that adhesion-independent durotaxis and hence frictotaxis, as shown here, is a widespread phenomenon that could be physiologically relevant for several cell types.

The theoretical model of asymmetric cell polarization on a friction gradient proposes that the side experiencing lower friction contracts faster, leading to myosin accumulation at the low-friction side, which then becomes the cell's rear. The authors could employ perhaps a form of traction force microscopy to provide evidence supporting this model.

We appreciate this suggestion. We agree with the reviewer that measuring traction forces during an event of cell polarization would be ideal for testing the mechanism proposed by our model. However, it is not possible as the tractions exerted by amoeboid cells are of about 1 Pa, which cannot be reliably measured with usual traction force microscopy (TFM) [Bergert et al. *Nat. Cell Biol.* **17**(4), 524-529, (2015)]. Moreover, using agar channels as a substrate for TFM presents further experimental challenges, such as the incorporation of beads into highly porous low-concentration agar gels. We hope that such a direct test becomes possible in the future.

The use of BSA to create friction gradients is also unclear and raises concerns about the potential of BSA to provide chemotactic or haptotactic signals that could influence cell migration.

As mentioned before (see comment to Reviewer 2), although the commercial name BSA contains the word 'serum', it solely contains albumin. BSA is purified albumin derived from blood serum, hence the name. BSA is commonly used in cell culture experiments and protein biochemistry/molecular biology research as a blocking agent to prevent unspecific protein binding. The most prominent known function of albumins is to maintain the oncotic blood pressure. To our knowledge, no studies report that BSA could induce a chemotactic response. Moreover, all our assays were carried out in medium without serum, preventing the possibility that serum proteins bind to BSA and eventually form a haptotactic gradient. To the best of our knowledge, the only integrin able to directly bind BSA (via a cryptic binding site) is α M β 2. This integrin, however, is only expressed on leukocytes and therefore does not interfere with our Frictotaxis assay. We added a comment about this point in the manuscript when we introduce the friction gradients made with BSA and PEG coating.

Reply to the reviewers

Reviewer #1 (Remarks to the Author):

The authors adequately answered my concerns and I can now recommend their work for publication in Nature Communications.

We thank the reviewer for recommending the acceptance of our manuscript in Nature Communications

Reviewer #2 (Remarks to the Author):

The authors have fully addressed all my concerns. I support publication of this manuscript in its current form.

We thank the reviewer for recommending the acceptance of our manuscript in Nature Communications

Reviewer #3 (Remarks to the Author):

After careful revision, the study "Frictiotaxis underlies focal adhesion-independent durotaxis" has been significantly improved, with images of adequate quality to support quantification and main conclusions. I believe that in its present form the manuscript is adequate for publication in Nature Communications.

We thank the reviewer for recommending the acceptance of our manuscript in Nature Communications

Reviewer #4 (Remarks to the Author):

The authors have provided a thoughtful response to the critiques, addressing the main concerns with new experimental data and clarifications. They effectively demonstrate that their agarose-based system does not support integrin-mediated adhesion, using various controls, including PLL-PEG coating and serum-free conditions. Additionally, they back up their argument by conducting extra experiments with integrin-blocking antibodies and inhibitors, showing that durotaxis efficiency is unaffected. The inclusion of HL60 cells, along with recent findings from Kang et al., further supports the physiological relevance of adhesion-independent migration, expanding the study's impact beyond the Walker 256 model.

However, the introduction of the term "frictiotaxis" still raises some questions. While the new data solidify the claim that the observed migration is adhesion-independent, a clearer distinction from other known forms of amoeboid motility would help. The authors should more explicitly contrast frictiotaxis with established forms of low-adhesion amoeboid migration to avoid overstating its novelty.

Overall, the manuscript has been significantly strengthened by the authors' responses, and the study presents high-quality data that contribute to the understanding of adhesion-independent migration. Refining the presentation of frictiotaxis as a novel concept and exploring its broader implications would enhance the work's impact.

We thank the reviewer for recommending the acceptance of our manuscript in Nature Communications. Following the reviewer's advice, we have made the distinction between our frictiotaxis model of migration and other amoeboid migration mechanisms based on low adhesion more explicit. In the Discussion, we state: "Although cells performing amoeboid migration might have weak adhesions, the mechanism of directed motion that we reveal is entirely based on friction. Accordingly, our model includes no adhesion, i.e. no resistance to normal pulling forces, but just friction that opposes cell-substrate sliding. This model is supported by our experimental observations in which suppression of integrin-based adhesions does not impair frictiotaxis. In a scenario in which weak adhesions were also present, durotaxis based on adhesion and friction could cooperate.**"**